# Molecular mechanism of activation-triggered subunit exchange in Ca²⁺/calmodulin-dependent protein kinase II

Moitrayee Bhattacharyya[1,2,3†], Margaret M Stratton[1,2,3†‡], Catherine C Going[4†], Ethan D McSpadden[1,2,3], Yongjian Huang[1,2,3,5], Anna C Susa[4], Anna Elleman[1,2,3§], Yumeng Melody Cao[1,2,3¶], Nishant Pappireddi[1,2,3], Pawel Burkhardt[1,3**], Christine L Gee[1,2,3], Tiago Barros[1,2,3], Howard Schulman[6], Evan R Williams[4], John Kuriyan[1,2,3,5,7*]

[1]Department of Molecular and Cell Biology, University of California, Berkeley, Berkeley, United States; [2]California Institute for Quantitative Biosciences, University of California, Berkeley, Berkeley, United States; [3]Howard Hughes Medical Institute, University of California, Berkeley, Berkeley, United States; [4]Department of Chemistry, University of California, Berkeley, Berkeley, United States; [5]Biophysics Graduate Group, University of California, Berkeley, Berkeley, United States; [6]Allosteros Therapeutics, Sunnyvale, United States; [7]Physical Biosciences Division, Lawrence Berkeley National Laboratory, Berkeley, United States

*For correspondence: kuriyan@berkeley.edu

†These authors contributed equally to this work

Present address: ‡Department of Biochemistry and Molecular Biology, University of Massachusetts at Amherst, Amherst, United States; §Department of Chemistry, Stanford University, Palo Alto, United States; ¶Department of Physics, Smith College, Northampton, United States; **Marine Biological Association, Plymouth, United Kingdom

**Abstract** Activation triggers the exchange of subunits in Ca²⁺/calmodulin-dependent protein kinase II (CaMKII), an oligomeric enzyme that is critical for learning, memory, and cardiac function. The mechanism by which subunit exchange occurs remains elusive. We show that the human CaMKII holoenzyme exists in dodecameric and tetradecameric forms, and that the calmodulin (CaM)-binding element of CaMKII can bind to the hub of the holoenzyme and destabilize it to release dimers. The structures of CaMKII from two distantly diverged organisms suggest that the CaM-binding element of activated CaMKII acts as a wedge by docking at intersubunit interfaces in the hub. This converts the hub into a spiral form that can release or gain CaMKII dimers. Our data reveal a three-way competition for the CaM-binding element, whereby phosphorylation biases it towards the hub interface, away from the kinase domain and calmodulin, thus unlocking the ability of activated CaMKII holoenzymes to exchange dimers with unactivated ones.

## Introduction

Ca²⁺/calmodulin-dependent protein kinase II (CaMKII) is a Ser/Thr kinase that is particularly important in neuronal signaling and cardiac function (*Colbran et al., 1989a*; *Giese and Mizuno, 2013*; *Hook and Means, 2001*; *Kandel et al., 2014*; *Lisman and Raghavachari, 2015*; *Lisman et al., 2002*; *Luo and Anderson, 2013*). CaMKII has a unique architecture, in which the catalytic domains are linked flexibly to a hub domain, also called the association domain, which forms a dodecameric or tetradecameric assembly (see *Figure 1A* for a description of the domains of CaMKII) (*Chao et al., 2011*; *Kanaseki et al., 1991*; *Kolb et al., 1998*; *Kolodziej et al., 2000*; *Morris and Török, 2001*; *Rosenberg et al., 2006*; *Stratton et al., 2013*; *Woodgett et al., 1983*). This organization, in which twelve or more kinase domains are maintained in close proximity, enables the highly cooperative activation of CaMKII by Ca²⁺/calmodulin (Ca²⁺/CaM) and the integration of calcium inputs (*Bradshaw et al., 2003*; *Chao et al., 2010, 2011*; *De Koninck and Schulman, 1998*).

**eLife digest** How does memory outlast the lifetime of the molecules that encode it? One enzyme that is found in neurons and has been suggested to help long-term memories to form is called CaMKII. Each CaMKII assembly is typically composed of 12 to 14 protein subunits associated in a ring and can exist in either an "unactivated" or "activated" state. In 2014, researchers showed that CaMKII assemblies can exchange subunits with each other. Importantly, an active CaMKII can mix with an unactivated CaMKII and share its activation state. CaMKII may use this mechanism to spread information to the next generation of proteins – thereby allowing activation to outlast the lifespan of the initially activated proteins. However the molecular mechanism that underlies this process was not clear.

Now, Bhattacharyya et al. – including some of the researchers involved in the 2014 work – address two questions about this mechanism. How do subunits exchange between CaMKII assemblies? And how does the activation of CaMKII initiate subunit exchange?

A closed-ring hub ties the subunits of CaMKII together, similar to the organization of the segments in an orange. To undergo subunit exchange, the hub must open up to release and accept subunits. Bhattacharyya et al. have now uncovered an intrinsic flexibility in the hub that is triggered by a short peptide segment in CaMKII. This segment, which is exposed in activated CaMKII but not in the unactivated form, can crack open the hub ring by binding between the hub subunits, like a finger separating the segments of an orange. This allows the hub to flex and expand, and once open, the hub's flexibility allows room for subunits to be released or accepted.

Although this subunit exchange mechanism could be a powerful means for spreading the activated state throughout signaling pathways, the biological relevance of this phenomenon has not been clarified. However, the mechanistic framework provided by Bhattacharyya et al. may allow new experiments to be performed that test the consequences of subunit exchange in live cells and organisms. It could also enable investigations into the importance of subunit exchange in long-term memory.

The activity of CaMKII is regulated by autophosphorylation (*Lai et al., 1986*; *Lou et al., 1986*; *Miller and Kennedy, 1986*; *Schworer et al., 1986*). In most kinases, phosphorylation of an "activation loop" that is located at the active site stabilizes the active conformation of the catalytic domain (*Huse and Kuriyan, 2002*; *Johnson et al., 1996*; *Taylor et al., 1992*). The activation loop of the CaMKII kinase domain has no phosphorylation site, and is in an active conformation (*Bulleit et al., 1988*; *Hanley et al., 1987*; *Lin et al., 1987*; *Rosenberg et al., 2005*). The three principal sites of autophosphorylation in CaMKII are located within the autoinhibitory segment, which follows the N-terminal kinase domain and blocks the catalytic site in the unactivated state of the enzyme (*Figure 1*).

One of the autophosphorylation sites, Thr 286 in the α isoform of human CaMKII (CaMKII-α), prevents rebinding of the autoinhibitory segment to the kinase domain upon phosphorylation. Phosphorylation of Thr 286 is the key step by which CaMKII gains calcium-independence (autonomy) after stimulation by $Ca^{2+}$/CaM (*Colbran et al., 1989b*; *Lou and Schulman, 1989*; *Miller et al., 1988*; *Schworer et al., 1988*; *Thiel et al., 1988*; *Waldmann et al., 1990*) (*Figure 1B*). The other two sites, Thr 305 and Thr 306, are located within an 18 residue CaM-binding element that is part of the autoinhibitory segment (*Figure 1B*) (*Meador et al., 1993*; *Rellos et al., 2010*). Phosphorylation of these two sites prevents the rebinding of $Ca^{2+}$/CaM to CaMKII (*Elgersma et al., 2002*; *Hanson et al., 1994*; *Hashimoto et al., 1987*; *Lickteig et al., 1988*; *Lou and Schulman, 1989*; *Mukherji and Soderling, 1994*). The importance of phosphorylation at these three sites is underscored by the fact that mice in which these sites are mutated exhibit distinct defects in learning and memory (*Elgersma et al., 2002*; *Giese et al., 1998*; *Silva et al., 1992*; *Wayman et al., 2008*).

We reported recently that the activation of CaMKII-α by ATP and $Ca^{2+}$/CaM triggers the exchange of activated subunits between different holoenzyme assemblies (*Stratton et al., 2014*). The replacement of Thr 286 by aspartate, which confers constitutive activity on the kinase (*Waldmann et al., 1990*), results in robust subunit exchange without $Ca^{2+}$/CaM, and the spread of

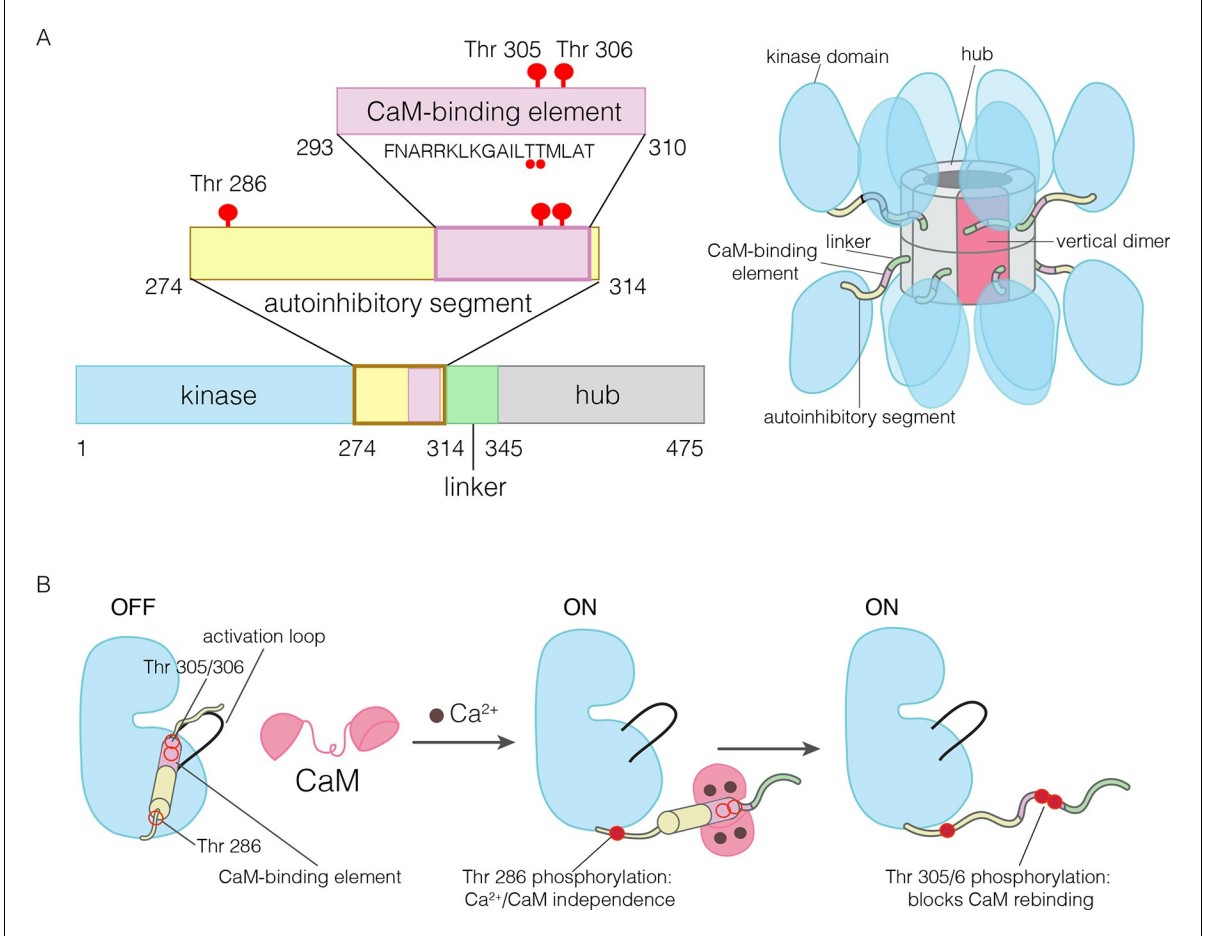

**Figure 1.** The domains of human CaMKII-α. (**A**) The architecture of the CaMKII holoenzyme is shown on the right. The hub domain forms a dodecamer or tetradecamer, and the kinase domains are flexibly linked to it by the autoinhibitory segment. (**B**) The autoinhibitory segment contains three critical sites of phosphorylation: Thr 286, Thr 305 and Thr 306. Activation by $Ca^{2+}$/CaM results in phosphorylation at Thr 286, which renders the enzyme independent of $Ca^{2+}$/CaM. Further phosphorylation of Thr 305 and Thr 306 prevents re-binding of $Ca^{2+}$/CaM.

activation to unactivated subunits. Our observation of subunit exchange connects with previous speculation that such a mechanism might allow maintenance of some level of activated CaMKII long after withdrawal of the activating stimulus (*Crick, 1984*; *Lisman, 1985*; *Lisman and Goldring, 1988*; *Lisman and Raghavachari, 2015*; *Miller and Kennedy, 1986*). It might also provide a mechanism to potentiate the effects of calcium stimuli under conditions where $Ca^{2+}$/CaM is limiting with respect to CaMKII, as in the dendritic spine (*Pepke et al., 2010*; *Persechini and Stemmer, 2002*).

We now present the results of experiments that clarify the molecular mechanism of activation-triggered subunit exchange in CaMKII. The two principal isoforms of CaMKII in the brain are the α and β isoforms, and we show that the β isoform also undergoes activation-dependent subunit exchange. We show that the intact human CaMKII-α holoenzyme exists as a mixture of dodecameric and tetradecameric forms, an observation that provides a link to a possible mechanism for subunit exchange. We demonstrate that the CaM-binding element of CaMKII is subject to a three-way competition for binding, between the kinase domain, $Ca^{2+}$/CaM and the hub. A structural understanding of how the CaM-binding element interacts with the hub and weakens its integrity emerged, unexpectedly, from studies on CaMKII homologs found in two distantly diverged species, the sea anemone *Nematostella vectensis* and the choanoflagellate *Salpingoeca rosetta*. These structures suggest that the docking of peptide segments at the interfaces between hubs can break the integrity of the ring.

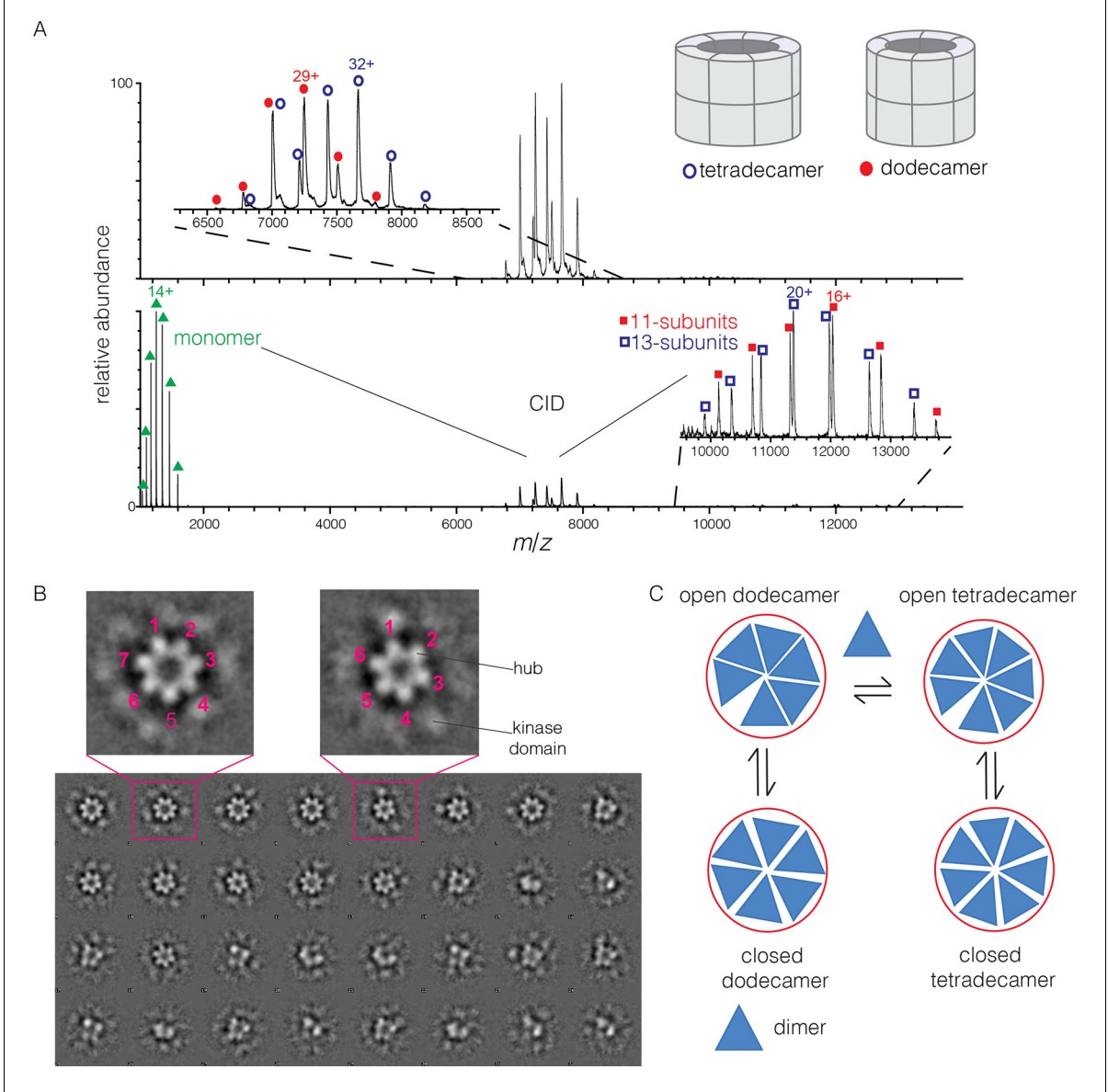

**Figure 2.** Human CaMKII-α forms both dodecamers and tetradecamers. (**A**) Native ESI-MS spectra of the human CaMKII-α hub reveals a ~1:1 mixture of dodecamers and tetradecamers (top panel). Collision-induced dissociation (CID) MS/MS of hub parent ions shows the presence of fragment ions corresponding to a highly charged hub monomer and a mixture of 11-subunit and 13-subunit species, clearly showing the mixed stoichiometry of the parent ion (bottom panel). (**B**) Two-dimensional class averages of images from negative-stain electron microscopy for human CaMKII-α show the existence of holoenzyme particles with both six-fold and seven fold symmetry. A subset of 32 of the 50 total class averages is shown, and an expanded view of two class averages is shown, one with seven-fold and one with six-fold symmetry. (**C**) A possible pathway for the transition between the dodecameric and tetradecameric species through the addition or loss of a dimer unit.

## Results and discussion

### The human CaMKII-α holoenzyme forms both tetradecameric and dodecameric assemblies

CaMKII is widely described as a dodecameric assembly, with six-fold cross-sectional symmetry. This view is based on negative-stain electron microscopic analyses (*Gaertner et al., 2004*; *Kolodziej et al., 2000*; *Morris and Török, 2001*; *Woodgett et al., 1983*) and was reinforced by a crystal structure of an intact dodecameric holoenzyme (*Chao et al., 2011*). It is therefore puzzling

that crystal structures of the isolated hub domains of CaMKII from *C. elegans* and mammalian species show them to be assembled into both dodecamers (*Rellos et al., 2010*) and tetradecamers (*Hoelz et al., 2003*; *Rosenberg et al., 2006*).

To determine the stoichiometry of hub assemblies in solution, we analyzed the human CaMKII-α hub domain by native electrospray ionization mass spectrometry (ESI-MS) (*Chowdhury et al., 1990*; *Heck, 2008*; *Sharon and Robinson, 2007*). The mass spectra demonstrate that the isolated hub assembly exists as a ~1:1 mixture of dodecamers and tetradecamers in solution. Collision-induced dissociation (CID) MS/MS of the mixture of hub parent ions (at 30 V collision energy) shows the presence of fragment ions corresponding to a hub monomer and a mixture of 11-subunit and 13-subunit species (*Figure 2A*). Collisional activation of intact gaseous protein complexes typically results in asymmetric dissociation, in which loss of a highly charged monomer subunit occurs as a result of structural changes and charge partitioning in the activated complex (*Jurchen and Williams, 2003*). This validates the mixed stoichiometry of the parent ion. Thus, the crystal structures of dodecameric and tetradecameric hubs are not artifacts of crystallization, but reflect instead a natural variation in the stoichiometry of assembly of the hub.

Due to poor signal in ESI-MS of full-length CaMKII holoenzyme, we examined unactivated human CaMKII-α holoenzyme by negative-stain electron microscopy (EM) in order to determine its stoichiometry, as described in Methods. An important aspect of our analysis is that no symmetry was imposed on the particles at any stage of the generation of class averages. The hub assemblies are clearly resolved in the EM micrographs, but the kinase domains are not, as is common for CaMKII. Visual inspection of the two-dimensional class averages clearly reveals a population of holoenzyme particles with seven-fold symmetry, in addition to those with the expected six-fold symmetry. It is unclear why particles with seven-fold symmetry were not reported in the previous EM analyses of CaMKII-α, which focused on dodecameric species (*Kolodziej et al., 2000*; *Morris and Török, 2001*). In our analysis, we could discern clear evidence, by visual inspection, for either six-fold or seven-fold symmetry in seven out of 50 classes each (the symmetry of the other class averages was not obvious). Based on the number of particles contributing to each of these 14 classes, we estimate the ratio of dodecameric to tetradecameric species to be roughly 55:45 (*Figure 2B*).

The observation that full-length human CaMKII-α exists in both dodecameric and tetradecameric forms has guided our thinking about how the exchange process might occur. If CaMKII dimers are the unit of exchange, as we hypothesized previously, then the dodecameric and tetradecameric species can interconvert by releasing and capturing dimers (*Figure 2C*). The release of dimers, rather than monomers, is also potentially significant for the maintenance of autonomous (Ca$^{2+}$/CaM independent) activity. Phosphorylation of Thr 286 can only occur *in trans* (*Hanson et al., 1994*; *Rich and Schulman, 1998*), and a dimeric unit may be able to maintain this phosphorylation whereas a monomer could not.

## The CaM-binding element of CaMKII binds to the hub with micromolar affinity

To analyze the mechanism of subunit exchange, we focused initially on potential interactions between the CaM-binding element and the hub. We had shown previously that mutation of the CaM-binding element blocks subunit exchange, and that phosphorylation of Thr 305 and Thr 306 in the CaM-binding element potentiates exchange (*Stratton et al., 2014*). We prepared both phosphorylated and unphosphorylated forms of peptides spanning the CaM-binding element of CaMKII-α (see Materials and methods). The peptides were labeled with a fluorophore (Bodipy-FL maleimide) at the C-terminal end, and binding was monitored by changes in fluorescence polarization (*Figure 3A*). To a fixed volume of labeled peptide (2 nM), increasing concentrations of the hub was added, and the change in fluorescence polarization was monitored (*Figure 3B*). This titration showed evidence for some degree of non-saturable binding at high hub concentrations. We therefore used a competition assay to determine the peptide affinities for the hub.

For the competition experiments, 2 nM of various labeled peptides were bound to 30 μM of the hub, resulting in high fluorescence polarization. Upon titration with the corresponding unlabeled versions of the peptides, the labeled peptides were competed off, and the fluorescence polarization decreased to the baseline value (*Figure 3C–E*). The value of the inhibition constant, $IC_{50}$, is ~100 μM for the unlabeled peptides, whether phosphorylated or not. The values of the corresponding dissociation constants ($K_D$), derived from non-linear least squares fit of the data by assuming that

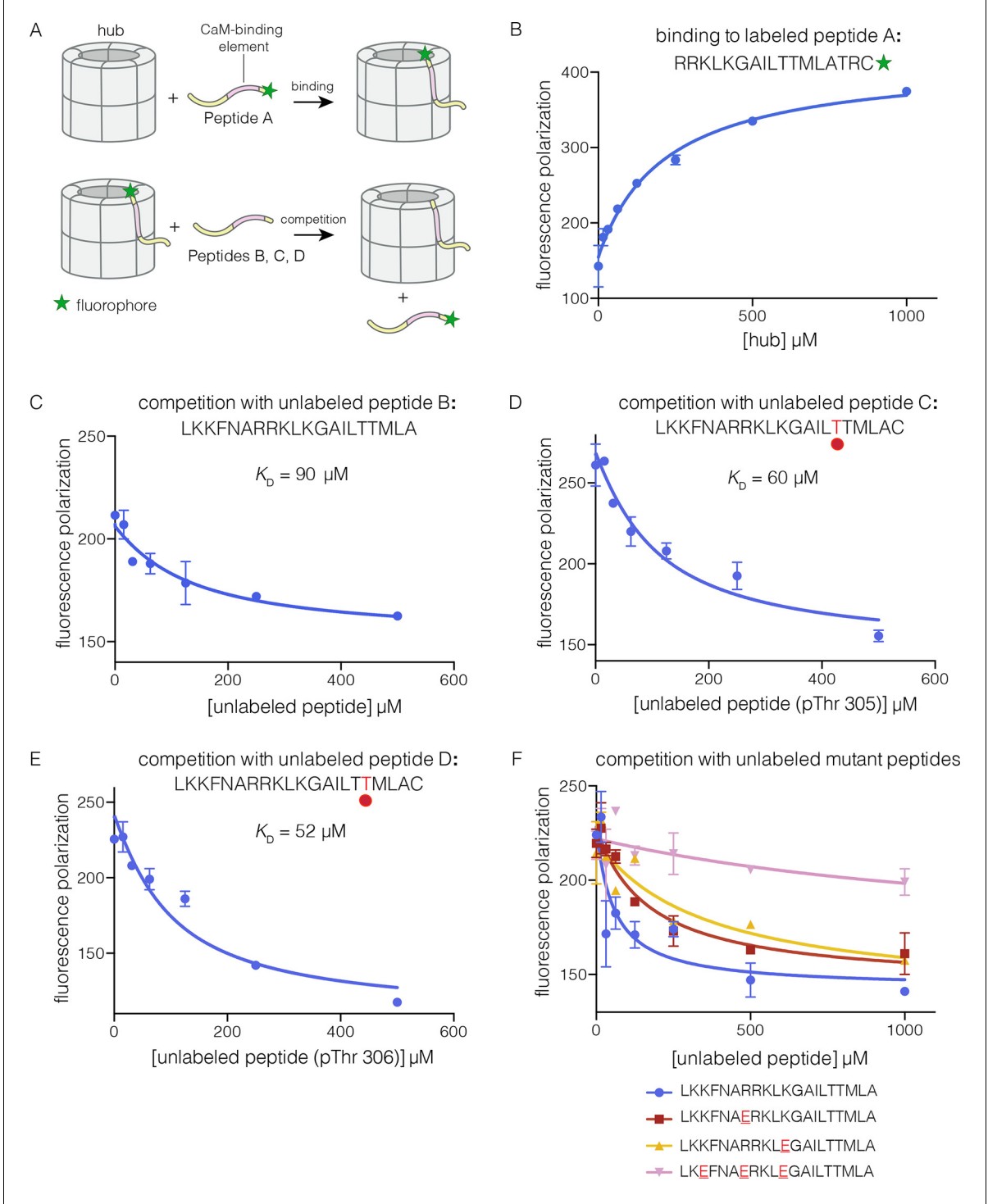

**Figure 3.** Binding of the CaM-binding element to the human CaMKII-α hub. (**A**) Schematic diagram of the binding experiments. (**B**) Direct titration of the hub to the labeled peptide A. Fluorescence polarization of the labeled peptide increases as a function of total hub concentration. The value of $K_D$ derived from these data is ~100 μm, but due to evidence for non-saturable binding, we relied instead on competition experiments to obtain $K_D$ values. (**C–E**) Competition experiments with unlabeled peptides. Addition of labeled peptide is followed by competition with excess unlabeled peptide of the same sequence (except for the competition with peptide B, where labeled Peptide A was used for initial binding). (**F**) Competition with mutant peptides. The hub is bound to labeled peptide A and fluorescence polarization is measured as a function of the total concentration of unlabeled mutant peptides. The sites of mutation in these peptides are highlighted in red and underlined. Fluorescence polarization is a ratio involving light intensities that are parallel and perpendicular to the plane of linearly polarized excitation light and is therefore reported as a dimensionless number.

*Figure 3 continued on next page*

*Figure 3 continued*

The following figure supplement is available for figure 3:

**Figure supplement 1.** Calculation of effective concentration of the CaM-binding element by assuming that the linker restricts it to within 50 Å of the hub.

each subunit has a single peptide-binding site, are ~90 μM, ~60 μM and ~50 μM, for the unphosphorylated peptide, and peptides phosphorylated on Thr 305 and Thr 306, respectively (see Materials and methods). These data show that phosphorylation of the peptide at either Thr 305 or Thr 306, separately, does not alter the affinity for the hub substantially (see *Figure 3D–E*). We could not obtain the doubly phosphorylated peptide, due to difficulty in its synthesis. Peptides spanning the CaM-binding element are inhibitors of the CaMKII kinase domain, which prevented generation of the doubly phosphorylated species through enzymatic means.

Prominent grooves are formed at the vertical interfaces between adjacent hub domains, and these can dock peptides, as discussed below. The edges of these grooves are lined by four acidic residues, two on each side of the interface: Glu 355, Glu 359, Glu 390 and Asp 393 in human CaM-KII-α, using the numbering in Protein Databank (PDB) entry 1HKX. The CaM-binding element contains the sequence [291]KKFNARRKLK[300]. If this peptide is docked as an additional strand on the open β sheet presented by one of the subunits, then the positively charged residues of the peptide could interact with the negatively charged residues in the hub.

We carried out competition experiments using unlabeled peptides in which positively charged residues in the KKFNARRKLK motif in the CaM-binding element were replaced by negatively charged residues (i.e., the variant peptide sequences are KKFNA**E**RKLK, KKFNARRKL**E**, and K**E**FNA**E**RKL**E**; the mutated residues are underlined). The triple-mutant peptide fails to compete the binding of the labeled peptide (*Figure 3F*). The two peptides bearing mutations at single sites have diminished ability to compete, compared to the unlabeled peptides. The $IC_{50}$ values of these peptides are increased by factors of about 2 and 4, relative to the wild type peptide. A more rigorous identification of the hub residues that constitute the peptide binding groove awaits further analysis of hub mutants.

Although the CaM-binding element has low affinity for the hub when added as a separate peptide, its effective concentration with respect to the hub is very high in the intact holoenzyme. We estimated the local concentration of the CaM-binding element by assuming that the linker restricts it to within 50 Å of the hub. This corresponds to an effective concentration of ~3 mM (see *Figure 3—figure supplement 1*). Thus, the observed $K_D$ of ~100 μM would allow the CaM-binding element to successfully engage the hub in the holoenzyme.

## Deleting the kinase domain of CaMKII results in spontaneous subunit exchange

Our finding that the CaM-binding element binds to the hub raises two questions: what prevents this element from interacting with the hub in unactivated CaMKII holoenzymes, and what is the role of phosphorylation in gating this interaction? Although phosphorylation of the CaM-binding element is not required for binding to the hub, the CaM-binding element will be sequestered by either the kinase domain or by $Ca^{2+}$/CaM in the absence of phosphorylation, and thereby prevented from interacting with the hub.

The unphosphorylated autoinhibitory segment, which includes the CaM-binding element, binds to $Ca^{2+}$/CaM with sub-picomolar affinity (*Tse et al., 2007*). Thr 305 and Thr 306 are buried within this high affinity interface (*Crivici and Ikura, 1995*; *Meador et al., 1992*), and phosphorylation would therefore disrupt the interaction with $Ca^{2+}$/CaM. The autoinhibitory segment also binds to the kinase domain, which it inhibits with an $IC_{50}$ value of ~1 μM, when added *in trans* as a peptide (*Colbran et al., 1988*). Phosphorylation on Thr 286 increases the affinity of CaMKII for $Ca^{2+}$/CaM by more than 1000-fold, by fully or partially releasing the autoinhibitory segment from the kinase domain (*Meyer et al., 1992*). Thus, the kinase domain sequesters the autoinhibitory segment, including the CaM-binding element, in the absence of phosphorylation, reducing its effective concentration with respect to the hub.

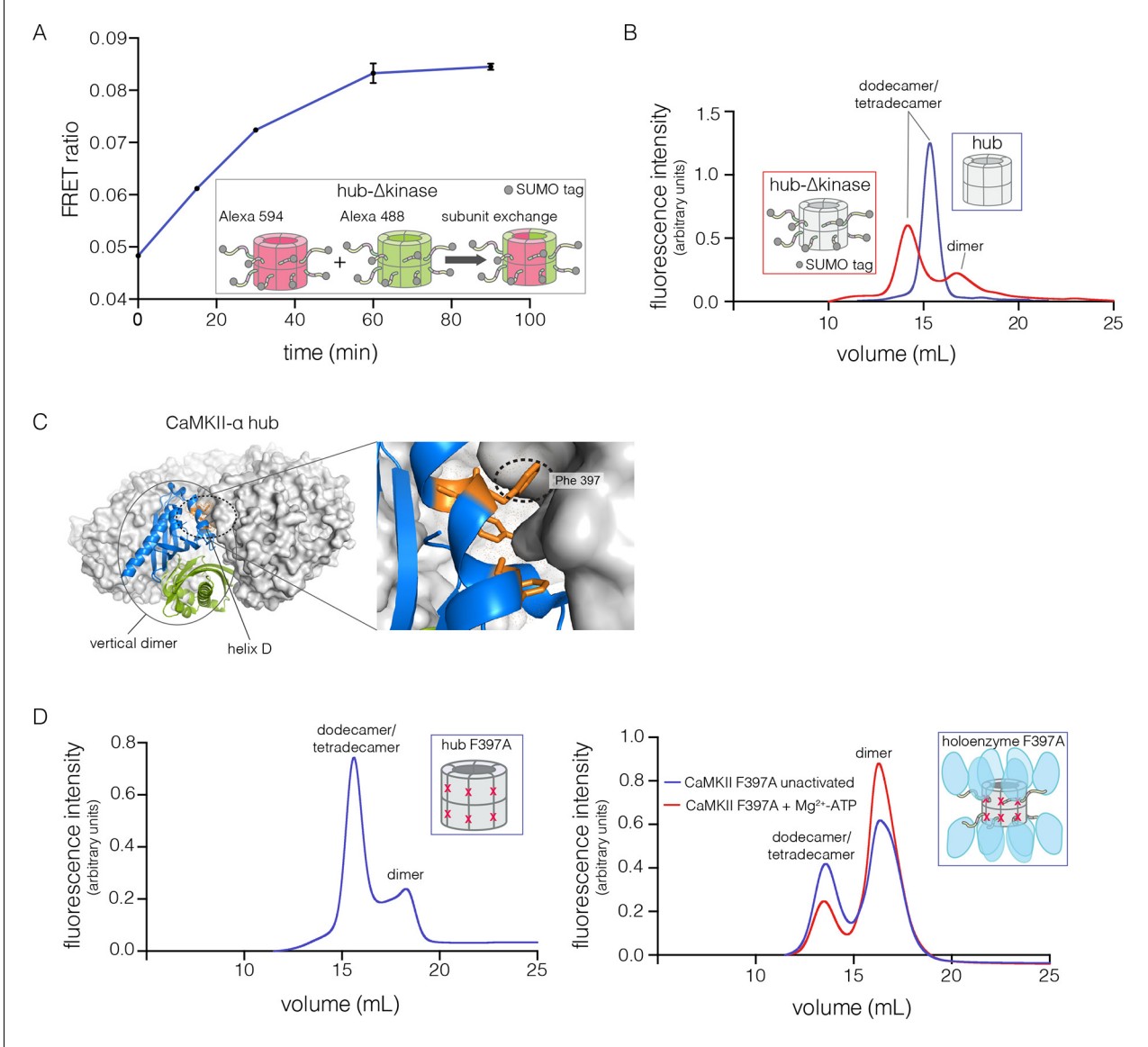

**Figure 4.** Analysis of kinase domain deletion and hub mutation. (**A**) Spontaneous subunit exchange in CaMKII lacking only the kinase domain (hub-△kinase). Increase in the FRET-ratio (see Materials and methods) is graphed as a function of time for the hub-△kinase construct (also see **Figure 4— figure supplement 1**). The increase in FRET upon mixing donor and acceptor-labeled proteins is indicative of subunit exchange (**Stratton et al., 2014**). (**B**) Analytical gel filtration of isolated hub domains and hub-△kinase from human CaMKII-α. The isolated hub shows one peak, with a retention volume corresponding to dodecamer/tetradecamer, as estimated from calibration data (see **Figure 4—figure supplement 4**). The hub-△kinase construct shows two peaks, corresponding to dodecamer/tetradecamer and dimer, respectively. Note that the hub-△kinase construct includes a SUMO tag, which accounts for its smaller retention volume compared to that for the hub alone. The gel filtration traces are normalized so that they have the same area under the curve. (**C**) A view of the lateral interface between subunits in adjacent vertical dimers of CaMKII-α (PDB code: 1HKX). (**D**) Replacement of Phe 397 at the hub interface by alanine leads to release of dimers from the isolated hub (left) and the holoenzyme (right). Pre-incubation of F397A holoenzyme with $Mg^{2+}$-ATP enhances the amount of dimers released, which is correlated with phosphorylation of Thr 305 and Thr 306 (data not shown). See also mass spectrometric analysis of the F397A mutant hub (**Figure 4—figure supplement 3**).

The following figure supplements are available for figure 4:

**Figure supplement 1.** Subunit exchange between various truncation constructs of CaMKII lacking the kinase domain (hub-△kinase), the kinase domain and the autoinhibitory segment (hub-linker) and the hub alone.

**Figure supplement 2.** Collection and re-injection of the dodecameric/tetradecameric peak for the human CaMKII-α F397A mutant re-releases dimers.

*Figure 4 continued on next page*

*Figure 4 continued*

**Figure supplement 3.** Mass spectra of 200 μM mutant hub in 1 M ammonium acetate (top), and 2 μM mutant hub in 25 mM Tris, 1 mM TCEP, pH 8 (middle) and 250 mM ammonium acetate, 25 mM Tris, 1 mM TCEP, pH 8 (bottom).

**Figure supplement 4.** Calibration curve for the analytical gel filtration column.

We tested the ability of the CaM-binding element to mediate subunit exchange when it is released from the kinase domain and Ca$^{2+}$/CaM. To do this, we made a construct comprised of the hub, the linker and the autoinhibitory segment of CaMKII-α, without the kinase domain (denoted hub-△kinase). The construct includes an N-terminal SUMO tag, which reduced proteolysis during purification. We monitored the gain of fluorescence resonance energy transfer (FRET) in order to test subunit exchange kinetics for this construct, by mixing samples labeled separately with donor and acceptor fluorophores, at a subunit concentration of ~5 μM, as described (*Stratton et al., 2014*). We observed that the hub-△kinase construct undergoes robust subunit exchange (*Figure 4A*). In contrast, the isolated hub of CaMKII-α, lacking the autoinhibitory segment and the linker, does not exchange subunits (*Stratton et al., 2014*). We also made a construct that spans the linker segment and the hub (residues 314 to 475, see *Figure 1A*; referred to as hub-linker). We observe robust subunit exchange only when the CaM-binding element is present in both samples (*Figure 4—figure supplement 1*). That is, a hub construct bearing the CaM-binding element cannot efficiently break a hub assembly lacking this element. We have shown previously that activated CaMKII holoenzyme, including the kinase domains, undergoes subunit exchange with unactivated holoenzymes (*Stratton et al., 2014*). The role of the kinase domains in destabilization of unactivated assemblies requires further study.

In order to check whether subunit exchange is accompanied by the release of subunits from the hub, we analyzed the hub assembly by analytical gel filtration (see Materials and methods). These experiments were typically done with an injection concentration corresponding to 700 nM CaMKII subunits; the concentration within the column itself is expected to be about ten-fold lower, based on the concentration of fractions recovered from the column. We monitored tryptophan fluorescence in order to obtain reliable protein signals at these low concentrations.

The isolated hub runs at a retention volume corresponding to a dodecamer or tetradecamer, showing no evidence for lower molecular weight species (*Figure 4B*). Strikingly, for the hub-△kinase construct, gel filtration reveals two peaks (*Figure 4B*). The major peak (~65% of the sample) corresponds to the dodecamer/tetradecamer, and the minor peak (~35% of the sample) corresponds to a dimeric species. This suggests that the integrity of the hub is compromised by the presence of the linker and the autoinhibitory segment. Reinjection of the eluate corresponding to the major peak results in two peaks at the same retention times as in the original experiment, suggesting that there is an equilibrium between dodecamers/tetradecamers and dimers (data not shown).

## The hub is assembled from vertical dimer units

The CaMKII hub consists of two rings that are joined together at the equatorial plane, each consisting of either six or seven subunits in dodecamers and tetradecamers, respectively. Given that the upper and lower rings do not separate readily (*Stratton et al., 2014*), we sought to check whether the hub would release monomers or "vertical dimers", comprising one subunit each from the upper and lower rings (see *Figure 1A*). Each subunit has three aromatic residues that form the core of the interface between adjacent subunits in the same ring (*Figure 4C*). We mutated one of these (Phe 397, the numbering is according to PDB code: 1HKX) to alanine in the isolated hub as well as in the holoenzyme, with the expectation that it would destabilize interactions between adjacent vertical dimers.

We then subjected the mutant proteins to analytical gel filtration. Both the mutant hub and holoenzyme exhibited a substantial release of dimers (~40% for the hub and ~60% for the holoenzyme), even at an elevated injection concentration of 3–5 μM (*Figure 4D*). Isolation and reinjection of the peak corresponding to the dodecamer/tetradecamer in the holoenzyme resulted in further release of dimers, suggesting that there was an equilibrium (*Figure 4—figure supplement 2*). Mass spectrometry of the F397A mutant hub confirmed the presence of dimers (*Figure 4—figure supplement 3*). This

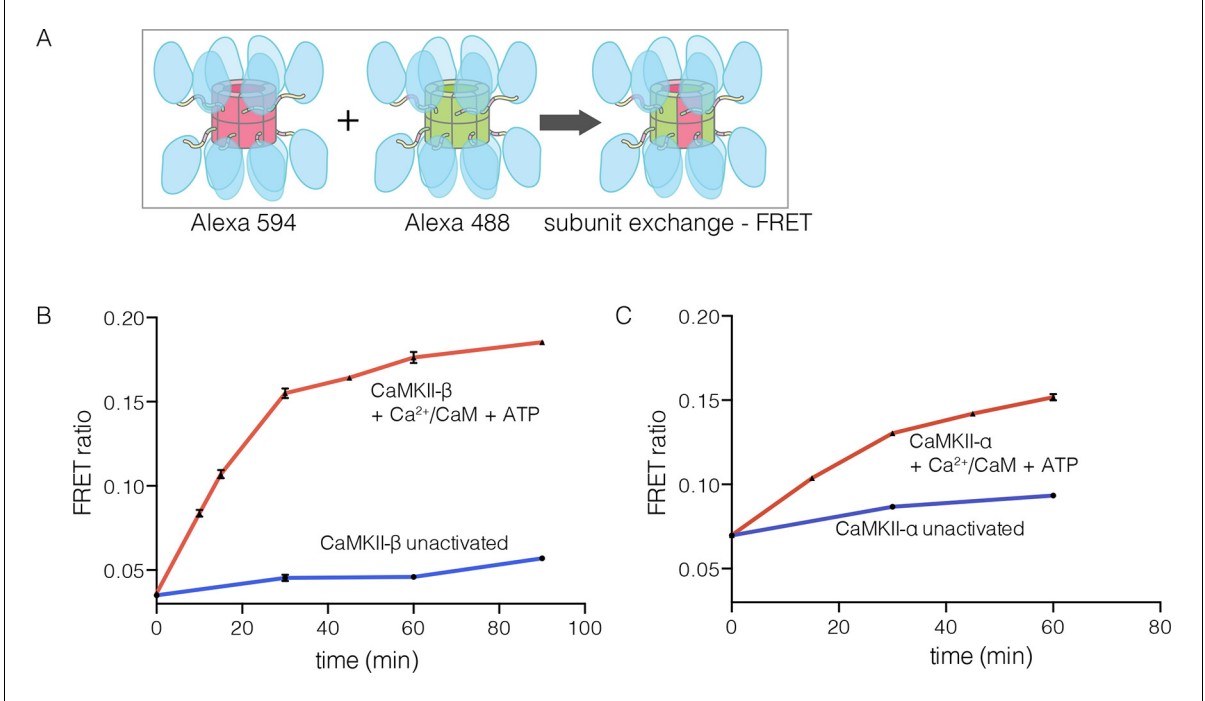

**Figure 5.** Subunit exchange in CaMKII-β. (A) Schematic diagram illustrating the design of solution FRET experiments for monitoring subunit exchange in CaMKII (*Stratton et al., 2014*). (B, C) FRET ratio as a function of time for human CaMKII-β and human CaMKII-α. The difference in the extent of FRET between the experiments shown in Panels B and C is due to differences in the percentage of labeled cysteine residues and the fluorophore environments in different proteins.

The following figure supplement is available for figure 5:

**Figure supplement 1.** Subunit exchange between human CaMKII-β and CaMKII-α.

is consistent with vertical dimers being the unit of assembly of the CaMKII hub, with each vertical dimer contributing one subunit each to the upper and lower rings.

## The β isoform of human CaMKII undergoes activation-triggered subunit exchange

CaMKII-α and CaMKII-β are the predominant species in the brain (*Tombes et al., 2003*). The principal difference between these two isoforms is the length of the linker, which is 218 residues long in CaMKII-β, compared to 30 residues in CaMKII-α. We purified CaMKII-β using a bacterial expression system, and tested its ability to undergo subunit exchange. We mixed proteins, labeled separately with donor and acceptor fluorophores and at a subunit concentration of ~5 μM, and monitored the development of FRET (*Figure 5A*). This experiment shows that CaMKII-β exchanges subunits, but only after stimulation with $Ca^{2+}$/CaM and ATP (*Figure 5B*). Thus, the phenomenon of activation-triggered subunit exchange is not limited to just the α isoform of CaMKII (*Figure 5B–C*). CaMKII-β can also exchange subunits with CaMKII-α in an activation-dependent manner, leading to the formation of CaMKII heterooligomers (*Figure 5—figure supplement 1*).

## Structures of human and *N. vectensis* CaMKII hubs reveal how peptides can open a closed hub

We now describe three new structures of hub assemblies that, together, provide support for the hypothesis that the CaM-binding element docks at the interfaces between vertical dimers, and suggest how such docking might destabilize and open the closed-ring assembly of the hub. The first insight came from a crystal structure of a dodecameric human CaMKII-α hub, for which the crystallization construct contained an additional 22 residues from an uncleaved expression tag. The tag

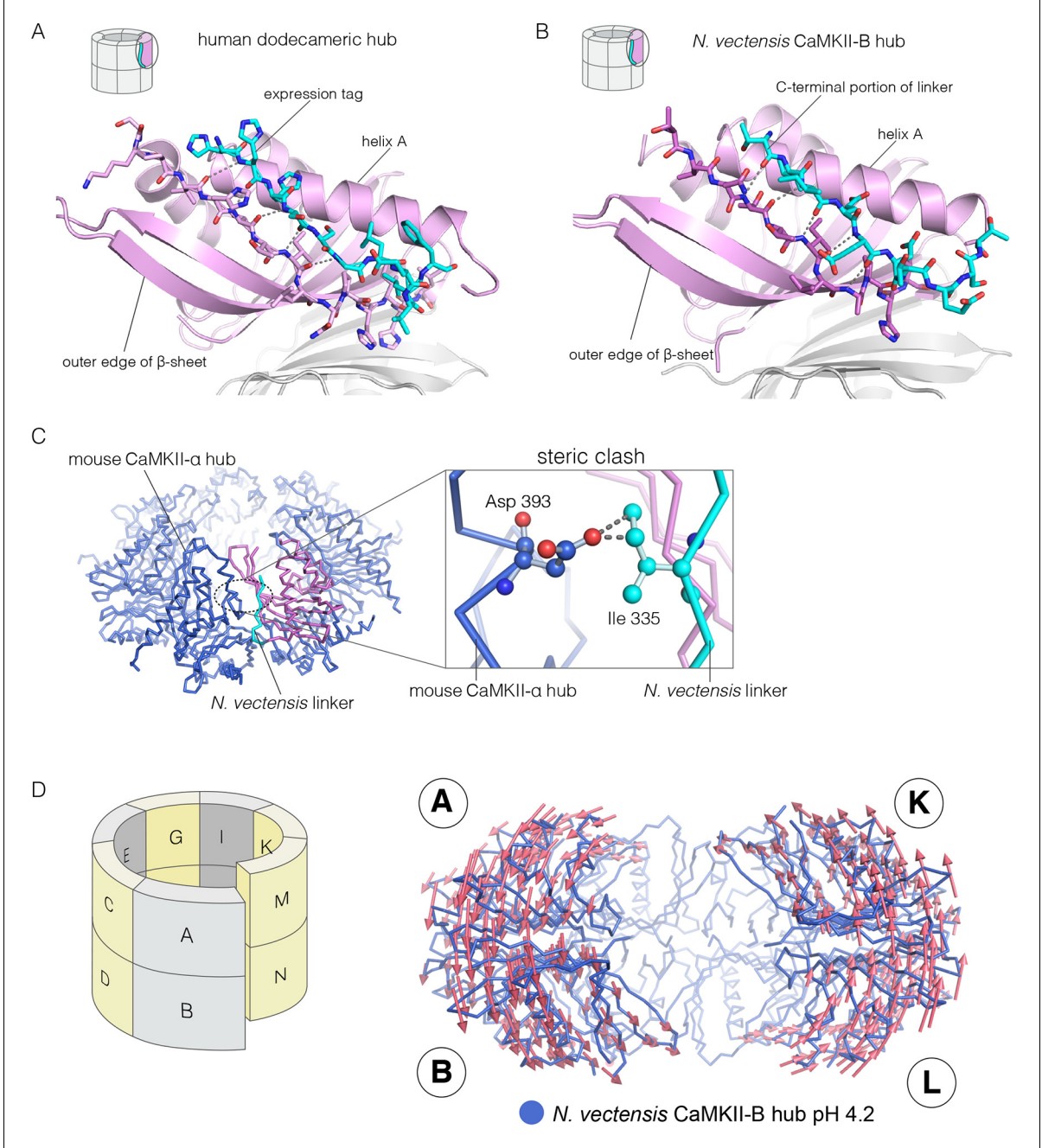

**Figure 6.** Docking of peptide segments onto the central β-sheet of the hub domain. (**A**) The N-terminal expression tag in human CaMKII-α docked on the open β-strand of the central β-sheet within the same hub subunit. (**B**) Linker residues in *N. vectensis* CaMKII-B hub (at pH 4.2) docked on the β-sheet of the hub subunit. (**C**) One subunit of the *N. vectensis* CaMKII-B hub, including the docked linker segment, is superimposed on the tetradecameric mouse CaMKII-α hub (PDB code: 1HKX). The *N. vectensis* hub domain is in magenta with the docked linker shown in cyan, and the CaMKII-α hub is in blue. The expanded view shows that the sidechain of Ile 335 in the linker of *N. vectensis* CaMKII-B hub collides with Asp 393 from helix D in the adjacent subunit of the mouse tetradecameric hub. (**D**) The distortion in the structure of *N. vectensis* CaMKII-B hub at pH 4.2. The spiral arrangement of the subunits is shown in the schematic diagram on the left, with each subunit labeled. The deviation between the closed-ring structure of *N. vectensis* CaMKII-A (pH 7.0) and the spiral form of the *N. vectensis* CaMKII-B structure is illustrated on the right, with the two structures aligned using subunits E, F, G and H. Arrows indicate displacement of the Cα atoms in going from the closed-ring CaMKII-A hub structure to the opened-ring CaMKII-B hub. Each arrow is scaled by a factor of 2 and arrows are only shown for Cα displacements greater than or equal to 2 Å. The structure shown is that of *N. vectensis* CaMKII-B hub. The M and N subunits are not shown for clarity.

The following figure supplement is available for figure 6:

*Figure 6 continued on next page*

*Figure 6 continued*

**Figure supplement 1.** Crystallographic axis of two-fold symmetry runs through the middle of the tetramer formed by the E-F and G-H dimers in *N. vectensis* pH 4.2 structure (CaMKII-B hub).

contains an N-terminal hexahistidine sequence, followed by a PreScission protease cleavage site (G SSHHHHHHSSGLEVLFQGPHM) (*Waugh, 2011*).

To our surprise, we found well-resolved electron density corresponding to the expression tag at two out of the twelve intersubunit interfaces. There was partial density for the tag at four more inter-subunit interfaces. The tag binds the hub almost precisely as we had predicted the CaM-binding element would, and it extends the central β-sheet of the hub domain by forming an additional parallel strand (*Figure 6A*). This docking allows the C-terminal end of the tag to connect to the N-terminal end of the first α-helix in the hub domain proper. The mouse CaMKII-α hub, which is 100% identical in sequence over the region spanning the human hub, was crystallized previously as a tetradecamer, rather than the dodecamer seen here (PDB code: 1HKX) (*Hoelz et al., 2003*). The groove between adjacent subunits is necessarily narrower in the tetradecamer than in the dodecamer, because of the hinging apart of the subunits, and the presence of the expression tag could favor the dodecamer for this reason.

The role of peptide docking in destabilizing the hub emerged from structures of the hubs of two CaMKII isoforms, denoted CaMKII-A and CaMKII-B, present in the sea anemone *Nematostella vectensis* (*Miller and Ball, 2008*; *Putnam et al., 2007*). *N. vectensis*, a member of the phylum *Cnidaria*, is an early branching metazoan (*Nakanishi et al., 2012*). The sequence identity between the hub domains in the two *N. vectensis* CaMKII isoforms is 62%, and these are 55% and 53% identical to the hub domain of human CaMKII-α respectively. The hub domain of CaMKII-A crystallized at pH 7, while that of CaMKII-B crystallized at pH 4.2. In each case, the construct used is very similar to that used to crystallize the mouse CaMKII-α hub (PDB code: 1HKX), and includes the last 12 residues of the linker.

The N-terminal α helix of the CaMKII-B hub is preceded by a 12-residue segment of the variable linker, including several acidic residues. This portion of the linker is highly conserved in metazoan CaMKII sequences. This segment is either absent in other crystal structures, or forms an extension of the N-terminal helix. In the structure of the *N. vectensis* CaMKII-B hub at pH 4.2, this segment is ordered in every subunit, and it folds back to dock onto the central β-sheet of the same hub subunit, extending the β-sheet by one strand, as seen in the structure described earlier for human CaMKII-α (*Figure 6B*). The acidic sidechains in this segment are located close to the acidic residues in the hub. This particular configuration is unexpected, and is presumably possible only because some or all of the acidic residues are protonated at the low pH of crystallization.

Comparison of the *N. vectensis* CaMKII-B hub at pH 4.2 with that of tetradecameric or dodecameric hubs shows that the linker peptide cannot be accommodated within the grooves of the closed-ring forms without steric clashes. In particular, the sidechain of Ile 335 in the *N. vectensis* docked linker would collide with the Asp 393 in helix D in the adjacent subunit if the geometries of either the tetradecameric or dodecameric hubs were preserved (*Figure 6C*).

These clashes are relieved in the pH 4.2 *N. vectensis* structure by a small distortion of the geometry of the assembly, away from a planar-ring configuration and into left-handed spiral geometry. The assembly is distorted at each interface in a systematic way throughout. This leads to the formation of a spiral, similar to a "lock-washer", with a slight opening between the A-B and M-N dimers (subunit notation is indicated in *Figure 6D*, *Figure 6—figure supplement 1*). To illustrate this distortion from planarity, we aligned the pH 4.2 *N. vectensis* CaMKII-B hub structure onto the pH 7.0 CaMKII-A hub structure by superimposing the four subunits that are furthest from the break (E, F, G, and H). We then illustrate the deviation between the two structures with arrows that indicate the displacement of Cα atoms from the flat closed-ring structure (pH 7.0) to the slightly spiral, ring-opened, structure (pH 4.2; the lengths of the arrows are scaled by a factor of 2 for clarity; *Figure 6D*). This analysis reveals that the A-B dimer is displaced in a direction opposite to that of the M-N dimer. This dislocation results in rupture of the interface between these two dimers.

Apart from the rupture at the A-B/M-N interface, the rest of the CaMKII-B hub assembly is symmetrical. At each interface, helix D from one subunit packs against the β-sheet of the other subunit

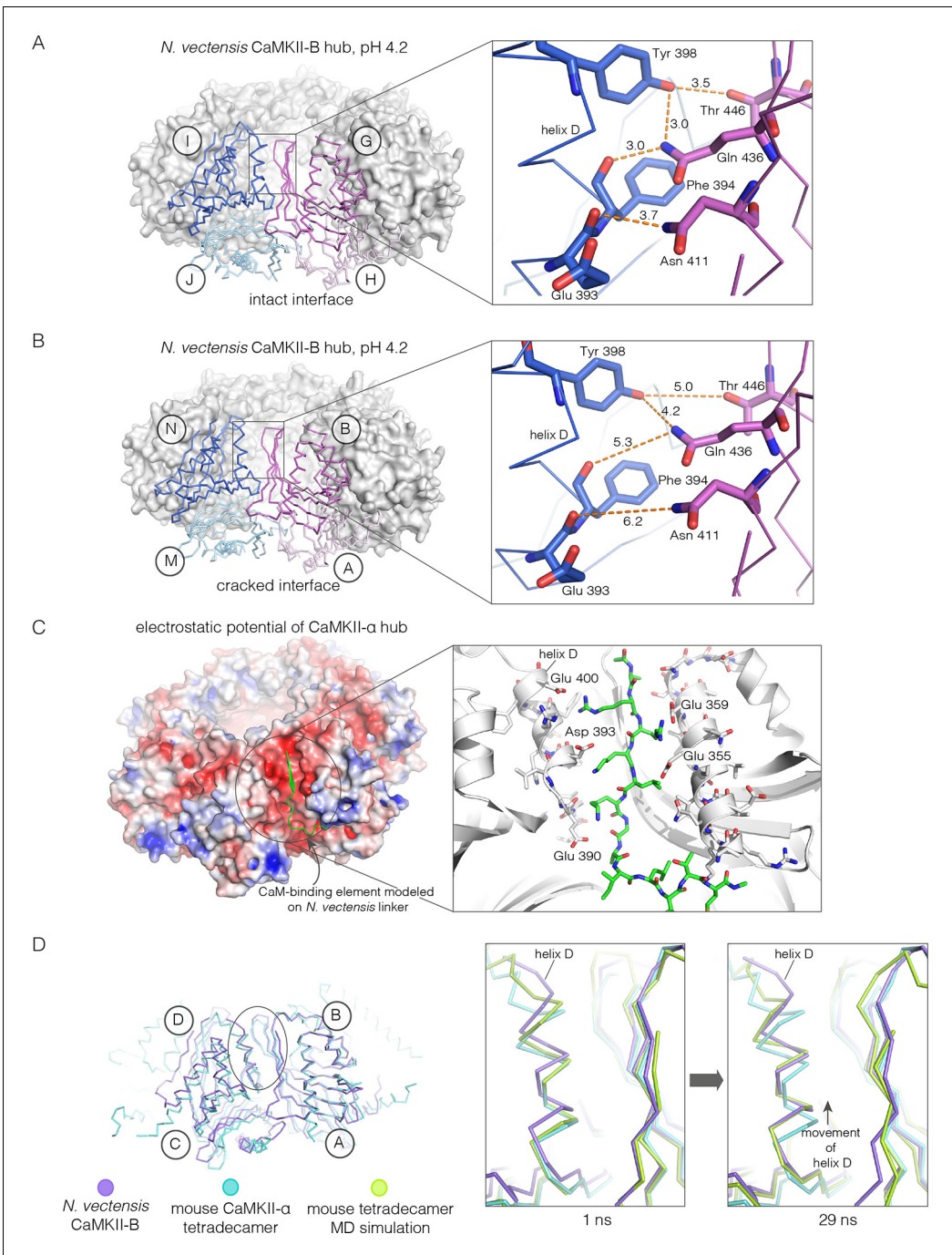

**Figure 7.** Analysis of the ring-opened *N. vectensis* CaMKII-B hub (pH 4.2) and its implications for docking of the CaM-binding element. (**A**) Interfacial hydrogen-bonding interactions at an intact interface in the *N. vectensis* CaMKII-B hub. (**B**) As in (**A**), but for one of the cracked interfaces. Hydrogen-bonding interactions are disrupted at this interface. (**C**) Modeling of the RRKLK motif of the CaM-binding element docked onto a hub interface in the tetradecameric mouse CaMKII-α hub (PDB code: 1HKX), based on the *N. vectensis* CaMKII-B hub structure. The surface electrostatic potential of the CaMKII-α hub, calculated using the Adaptive Poisson-Boltzmann Solver (APBS) in PyMOL, is shown, with red and blue representing negative and positive electrostatic potential, respectively. The expanded view shows interactions between the modeled CaM-binding element and residues in the hub. (**D**) Left, structural superposition of the closed-ring tetradecameric mouse CaMKII-α hub (PDB code: 1HKX) on the ring-opened *N. vectensis* CaMKII-B hub. The structures were aligned using subunit B. Note that the disposition of helix D is different at the two interfaces (circled). The expanded views show instantaneous structures (at 1 ns and 29 ns) from a molecular dynamics trajectory of the mouse hub, with the modeled CaM-binding

*Figure 7 continued on next page*

*Figure 7 continued*

element. Note that helix D in the simulation moves from its initial position to one that is closer to the position of this helix in the ring-opened *N. vectensis* structure.

(*Figure 7A*). A very similar interfacial interaction is seen in both the tetradecameric mouse CaMKII-α hub (PDB code: 1HKX) and the dodecameric human CaMKII-γ hub (*Rellos et al., 2010*) (PDB code: 2UX0). This interaction is ruptured at the A-B/M-N interface in *N. vectensis* CaMKII-B at pH 4.2 (*Figure 7B*). For example, at the interface between subunits B and N, the sidechains of Gln 436, Thr 446 and Asn 411 are pulled away from their hydrogen bonding partners in the other subunit.

The segment that is bound at the interfacial grooves in the ruptured *N. vectensis* structure is not the CaM-binding element, which was not part of the crystallization construct. Instead it is the C-terminal portion of the linker, which is presumably accommodated due to the low pH of crystallization. We modeled how the CaM-binding element might dock on the hub by aligning a hub subunit from the *N. vectensis* CaMKII-B structure on to subunits of dodecameric human CaMKII-γ (PDB code: 2UX0) and tetradecameric mouse CaMKII-α (PDB code: 1HKX). This allowed us to transfer the docked peptides on to the models for the closed-ring hubs, as shown in *Figure 7C* for the tetradecameric ring. We then altered the sequence of the peptide to an RRKLK motif that is present in the CaM-binding element. This results in good electrostatic complementarity between the peptide sidechains and acidic sidechains in the hub (*Figure 7C*).

We initiated molecular dynamics trajectories from these structures. For the tetradecameric simulations (two independent trajectories, 100 ns each), a conformational change occurs at the interface in both trajectories (*Figure 7D*); no corresponding change was seen in the dodecamer in these relatively short trajectories. Remarkably, the structural change closely resembles the difference seen between the ring-opened *N. vectensis* CaMKII-B hub and the closed tetradecameric rings in *N. vectensis* CaMKII-A and mouse CaMKII-α. Thus, we propose that electrostatic complementarity between the CaM-binding element and the hub interface allows the docking of this element onto the hub. Engagement of the interface by the CaM-binding element then results in a conformational change that favors a ring-opened "lock-washer" configuration of the assembly, rather than a closed ring.

## The hub domain of a choanoflagellate CaMKII forms a ring-opened spiral assembly

Choanoflagellates are the closest living relatives of metazoans (*Carr et al., 2008*; *King et al., 2008*; *Richter and King, 2013*). The choanoflagellate *S. rosetta* has one CaMKII gene (*Burkhardt et al., 2014*; *Fairclough et al., 2013*). The *S. rosetta* kinase domain, including the autoinhibitory segment, and the hub are 52% and 41% identical in sequence to the corresponding domains of human CaMKII-α. Although *S. rosetta* CaMKII contains a residue that is equivalent to Thr 286, it lacks the two inhibitory phosphorylation sites within the CaM-binding element of the autoinhibitory segment (Thr 305 and Thr 306), as well as sites of regulatory glycosylation, nitrosylation and oxidation (*Coultrap et al., 2014*; *Erickson et al., 2008*, *2013*, *2015*).

We expressed, purified, and crystallized the kinase and hub domains of *S. rosetta* CaMKII separately. The structure of the autoinhibited kinase domain was determined at 2.9 Å resolution, and is very similar to that of autoinhibited human CaMKII-δ kinase domain (*Rellos et al., 2010*) (PDB code: 2VN9; *Figure 8—figure supplement 1*).

The *S. rosetta* CaMKII hub forms a vertical dimer that closely resembles those formed by the hubs of mammalian (PDB codes: 1HKX and 2UX0), the nematode *C. elegans* (*Rosenberg et al., 2006*) (PDB code: 2F86) and *N. vectensis* CaMKII proteins. The mouse CaMKII-α hub dimer can be superimposed on the *S. rosetta* dimer with a rms deviation in Cα positions of 1.4 Å over 197 residues. Note, however, that the close overlap of the overall structure of the dimer masks an important conformational difference within each domain, which we discuss below.

The quaternary assembly of the *S. rosetta* hub is strikingly different from that of all other hubs in that it forms a right-handed spiral, corresponding to a hexagonal $P6_1$ screw axis in the crystal lattice, instead of a closed ring (*Figure 8A*). On looking at a projection down the hexagonal screw axis, the dimers are arrayed in a circle with almost the same diameter as that formed by the dodecameric

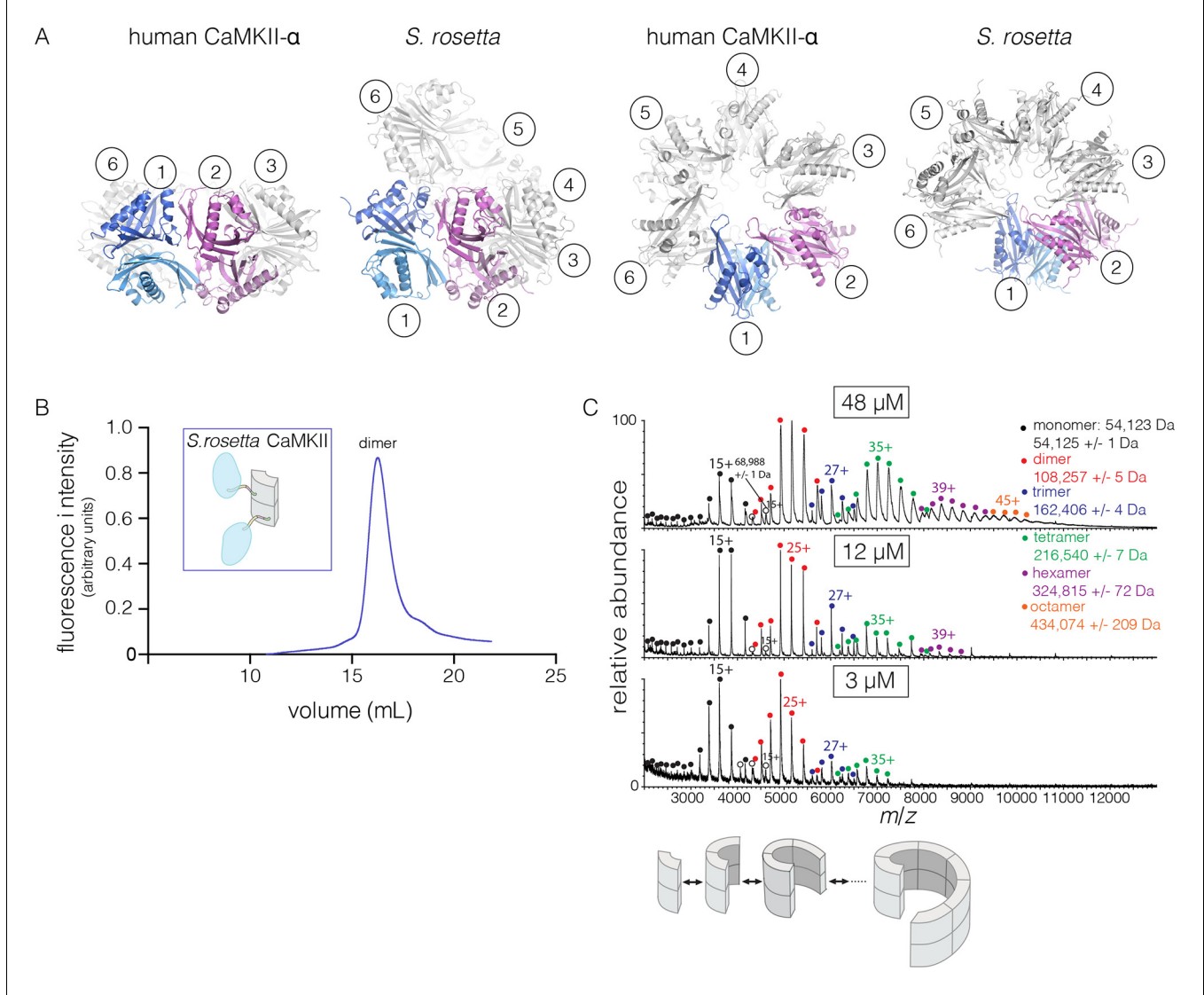

**Figure 8.** Structure and stoichiometry of the *S. rosetta* CaMKII. (**A**) Structural comparison of the hub assemblies of human CaMKII-α and *S. rosetta* CaMKII. Two views are shown for comparison and the six dimeric units are labeled 1–6. (**B**) Analytical gel filtration of *S. rosetta* CaMKII holoenzyme (blue) shows one predominant peak, corresponding to a dimeric species, as estimated from calibration data (see *Figure 4—figure supplement 4*). (**C**) Native electrospray-ionization mass spectrometry (ESI-MS) for the *S. rosetta* CaMKII holoenzyme at different concentrations. The proportion of higher order oligomeric species increases with increasing concentration, with an even number of subunits in each major species.

The following figure supplements are available for figure 8:

**Figure supplement 1.** (A) Structural superposition of the autoinhibited kinase domain from *S. rosetta* CaMKII on the autoinhibited human CaMKII-δ kinase domain (PDB code: 2VN9).

**Figure supplement 2.** Multi-angle light scattering (MALS) analysis at a concentration of ~200 μM reveals the existence of tetramers in *S. rosetta* CaMKII hub.

---

forms of human CaMKII. An orthogonal view shows that the helical pitch of the assembly is such that the sixth dimer is displaced vertically by the length of a dimer, with respect to the first one (*Figure 8A*). This positions the seventh dimer directly above the first one. The spiral formed by hub dimers continues in the crystal lattice.

## The *S. rosetta* CaMKII holoenzyme forms concentration-dependent oligomers that readily exchange dimers

The open-ended spiral assembly of *S. rosetta* hub dimers suggests that assemblies formed by the *S. rosetta* CaMKII holoenzyme would not possess defined stoichiometry. At an injection concentration of 700 nM, full-length *S. rosetta* CaMKII forms a dimer, as revealed by analytical gel filtration (*Figure 8B*). Multi-angle light scattering (MALS) analysis at much higher concentration (~200 µM) reveals the existence of tetramers (*Figure 8—figure supplement 2*).

We also examined the concentration dependence of the stoichiometry of *S. rosetta* CaMKII by native ESI-MS. In contrast to the human CaMKII-α holoenzyme, for which good mass spectral signal and resolution could not be obtained, full-length *S. rosetta* CaMKII yielded high quality data. At 3 µM, the mass spectrum reveals mainly monomers and dimers, and a low abundance of tetramers (*Figure 8C*). At 12 µM, the tetramer population increases, and a small population of hexamers also emerges. At 48 µM, the tetramer and hexamer populations increase further, and a population of octamers now appears. These results are consistent with the open-ended spiral assembly of the *S. rosetta* hub, which can gain or lose dimer units without steric constraints.

The variability in the stoichiometry of *S. rosetta* CaMKII suggests that it should undergo subunit exchange readily, without activation. We verified that this is the case by using the FRET-based

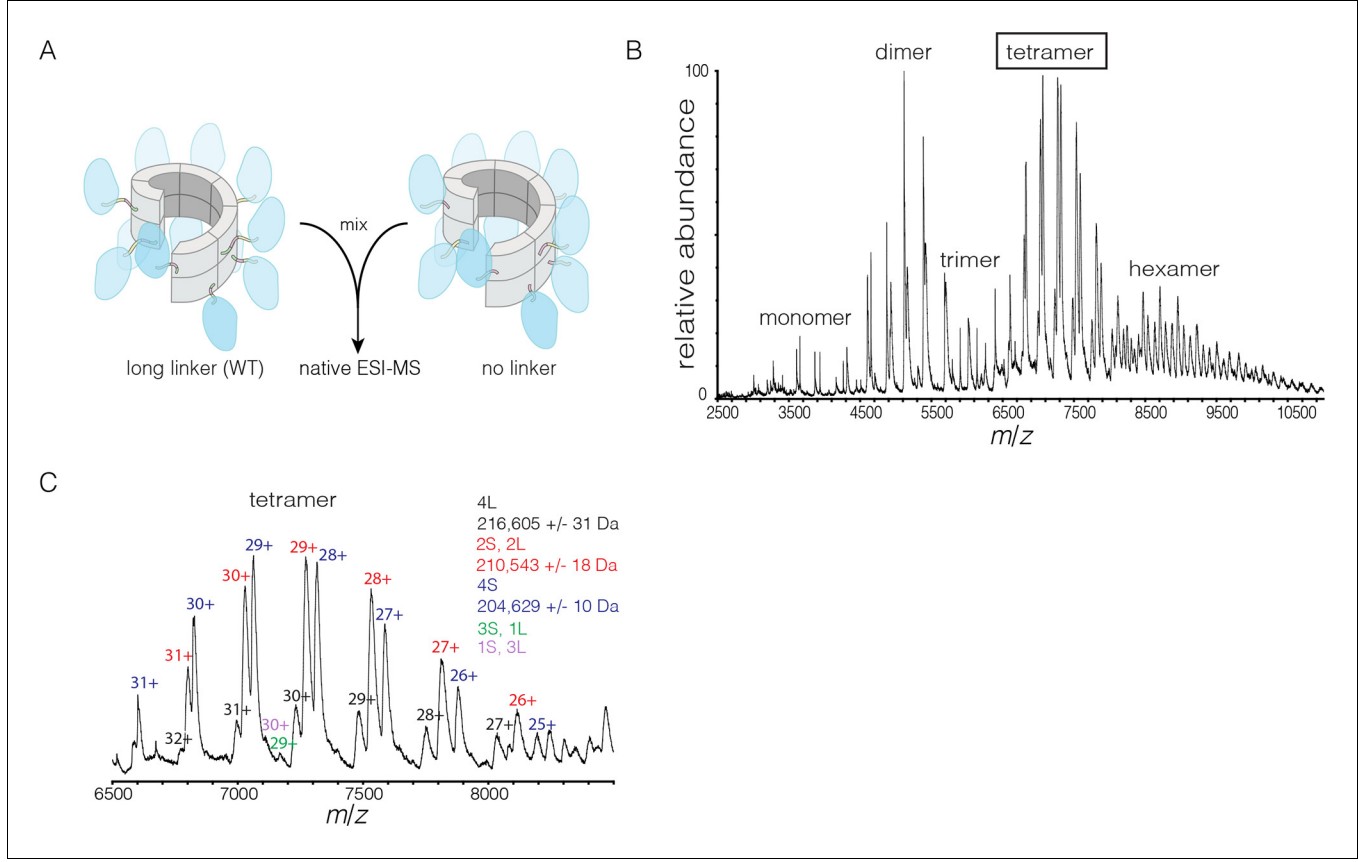

**Figure 9.** Subunit exchange in *S. rosetta* CaMKII monitored by ESI-MS. (**A**) Setup of the subunit exchange experiment for *S. rosetta* CaMKII. Wild-type CaMKII is mixed with a shorter version in which the linker is deleted, followed by mass spectrometric analysis. (**B**) Mass spectrum showing a mixture of multiple oligomeric species, ranging from monomer to hexamer. (**C**) Expansion of the mass spectrum in the region of the intact tetramer shows the existence of even numbers of long and short species. This indicates that subunit exchange in *S. rosetta* CaMKII occurs predominantly through the exchange of dimer units. 1L, 2L, 3L and 4L refer to CaMKII assemblies with one, two, three or four long (wild-type) subunits, respectively. Likewise, 1S, 2S, 3S and 4S refer to the corresponding numbers of short (no linker) subunits.

The following figure supplement is available for figure 9:

**Figure supplement 1.** Spontaneous subunit exchange in *S. rosetta* CaMKII without activation.

exchange assay. Separate samples of full length *S. rosetta* CaMKII were labeled with donor (Alexa-488) and acceptor (Alexa-594) fluorophores, respectively, followed by mixing at ∼5 µM final subunit concentration for each sample, without ATP or $Ca^{2+}$/CaM. This results in a steady increase in FRET, consistent with subunit exchange (*Figure 9—figure supplement 1*).

We also monitored subunit exchange using native ESI-MS, which revealed that exchange proceeds through dimer units. We engineered a variant of *S. rosetta* CaMKII in which the linker connecting the kinase domain to the hub is deleted. This shorter variant can now be distinguished from the wild type in mass spectra. For the exchange experiment, short (no linker) and long (wild type) forms of *S. rosetta* CaMKII are mixed at subunit concentrations of ∼280 µM each and incubated for 15 min at 37°C. This was followed by buffer exchange into 1 M ammonium acetate (neutral pH) and the spectrum was acquired at a final concentration of ∼10 µM (*Figure 9A*). This results in the generation of oligomers of variable stoichiometry, containing both short and long forms, in even stoichiometry (*Figure 9B*).

The mass spectra reveal that subunit exchange occurs exclusively through exchange of dimers. For example, examination of the peaks corresponding to dimeric and tetrameric assemblies shows that the mixed species are composed predominantly of even numbers of short and long subunits (data for tetrameric species are shown in *Figure 9C*). This is generally true for the hexameric species as well. Although a small amount of trimeric species is detected, they are composed predominantly of all short or all long subunits (data not shown). This suggests that the trimer represents a species that is not in equilibrium with other species.

## Normal mode calculations and molecular dynamics simulations indicate that the human CaMKII hub can adopt spiral conformations

Despite the striking dissimilarity in quaternary structure, the interface that holds dimers together in all metazoan CaMKII hubs is very closely preserved in the *S. rosetta* hub (*Figure 10A–C*), with helix D from one hub domain packed against the β-sheet of the adjacent domain. When an interfacial region in the mouse tetradecameric hub assembly is superimposed on a corresponding region in the *S. rosetta* hub, the rms deviation in Cα positions is 0.35 Å over 22 residues. The interfacial hydrogen bonding network is somewhat weakened in the *S. rosetta* hub assembly due to replacement of a tyrosine residue by a phenylalanine (*Figure 10C*).

We wondered how the CaMKII hub can form different assemblies while maintaining the interactions that hold vertical dimers together. To answer this, we compared the *S. rosetta* hub structure with that of other hubs (*Figure 10D*). Each hub domain consists of two layers: an inner layer that consists of a four-stranded antiparallel β-sheet and an outer layer formed by three α-helices, which precede the inner layer in sequence (two shorter additional strands of the β-sheet form a segment that intervenes between two of the α-helices). The principal structural difference between a subunit in a closed-ring hub and one in the spiral *S. rosetta* hub is a rotation in the outer edge of the β-sheet with respect to the inner edge of the sheet and the α-helices (see *Figure 10D*). The interfacial interactions between the vertical dimers of the hub are between residues on the outer edge of the β-sheet in one subunit, and residues presented by helix D in the other. These interfacial interactions are preserved in the *S. rosetta* hub, but a change in the curvature of the β-sheet results in a change in the orientation of helix D in each subunit. This results in a change in the quaternary assembly as the β-sheet/helix D interaction is propagated from subunit to subunit.

A variation in the curvature of the β-sheet explains the variation in the assembly of all the hubs we have examined: the dodecameric and tetradecameric closed-ring hubs, the slightly ruptured left-handed spiral of the *N. vectensis* CaMKII-B hub and the open right-handed spiral of the *S. rosetta* hub (see *Figure 10E* for a schematic illustration of how a change in curvature of the β-sheet explains the transition from the dodecameric to the tetradecameric closed-ring forms). In each case, as the extent of curvature of the β-sheet changes, the interfaces between the subunits are maintained while changing the geometry of the oligomeric assembly (as illustrated in *Figure 10E*). Importantly, the docking of a peptide between two subunits can force a change in the curvature of the β-sheet, thereby forcing a change in the oligomeric assembly.

Because the spiral *S. rosetta* structure so closely preserves the interactions at the interfaces, we wondered whether the human hub might be able to adopt a spiral form. We considered whether this is possible by carrying out normal mode analyses and molecular dynamics. We calculated the normal modes of a mouse CaMKII-α hub vertical dimer by using an elastic network model, as

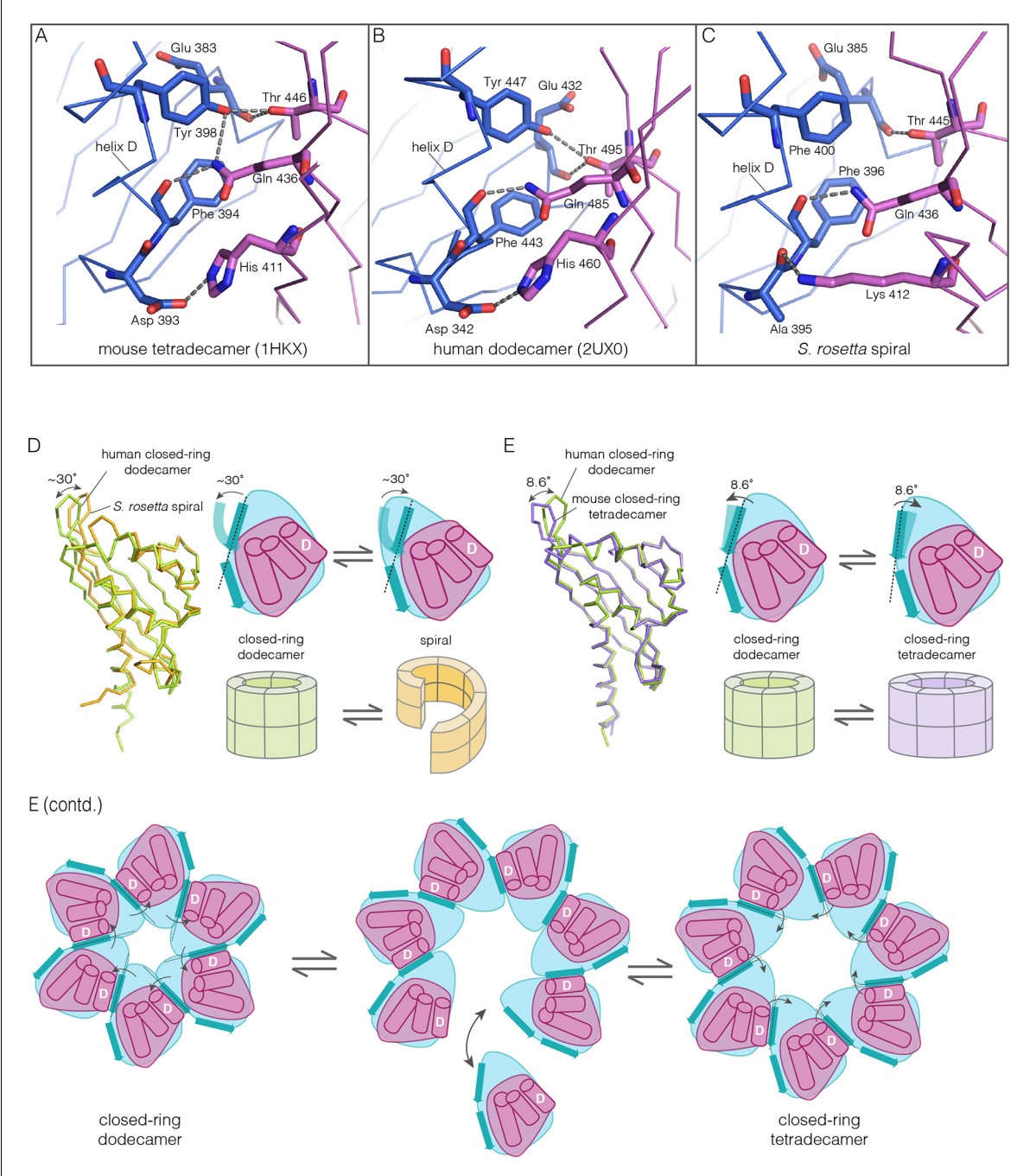

**Figure 10.** How the CaMKII hub accommodates assemblies with variable stoichiometry. (**A–C**) Comparison of interfacial interactions at the interface between vertical dimers in the mouse tetradecameric hub (PDB code: 1HKX), human dodecameric hub (PDB Code: 2UX0) and the *S. rosetta* hub respectively. (**D**) Structural comparison between a subunit of the *S. rosetta* spiral hub and a human dodecameric hub. The schematic diagram represents a hub domain in terms of two layers, one formed by the β-sheet and one by the α-helices. The conformational change in the hub in different assemblies corresponds to a change in the twist of the β-sheet. (**E**) Structural comparison of tetradecameric and dodecameric closed-ring hubs. Changes in the curvature of the β-sheet allow interfacial packing between adjacent subunits to be maintained in each case. The schematic illustration shows how a change in curvature of the β-sheet explains the transition from the dodecameric to the tetradecameric closed-ring forms, preserving the interfacial interactions.

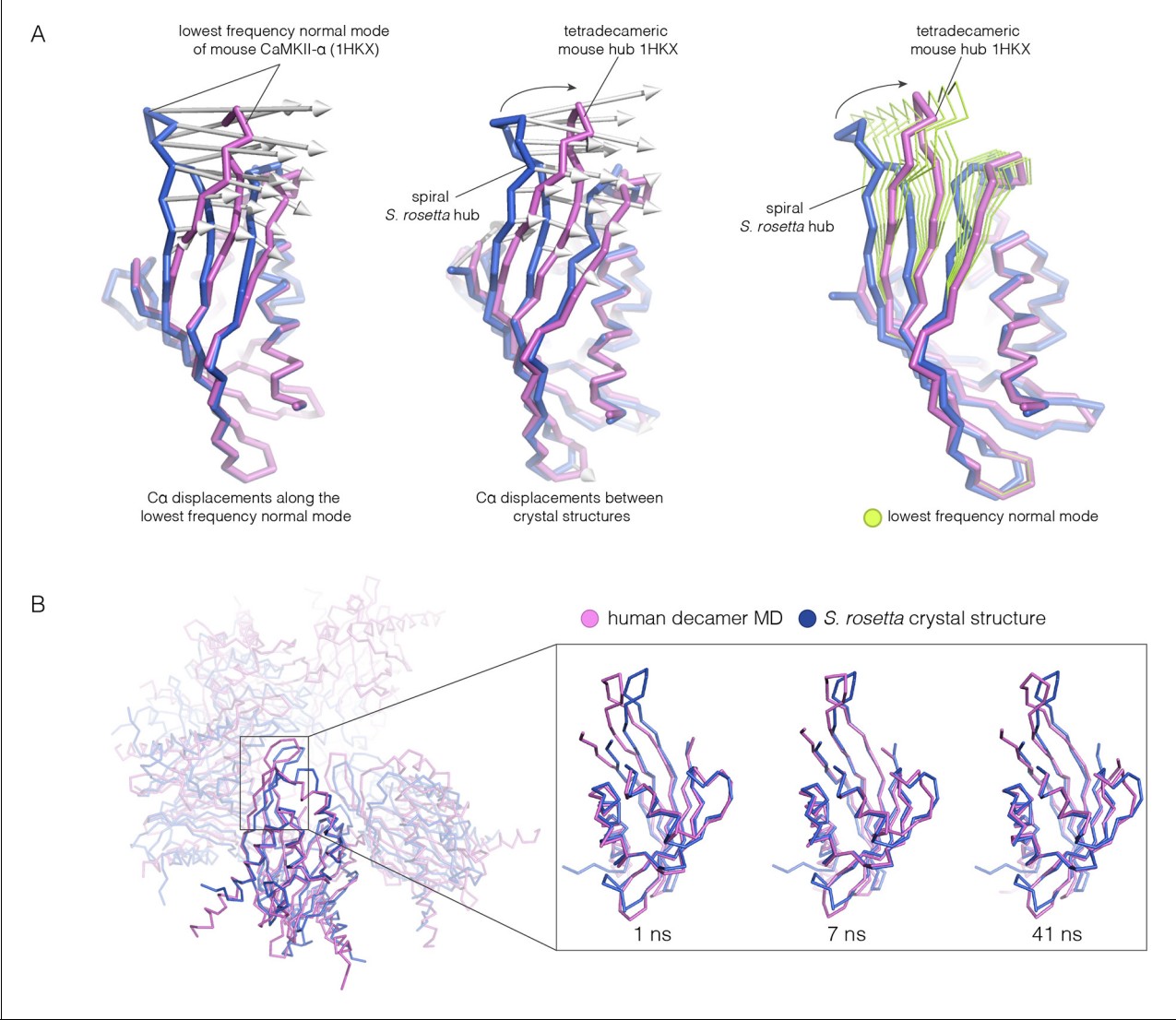

**Figure 11.** Normal mode calculation and molecular dynamics simulations for the mouse and human hub respectively. (**A**) Movement of the mouse CaMKII-α hub (PDB code: 1HKX) along the lowest-frequency internal normal mode recapitulates the structural differences in the *S. rosetta* hub. Normal modes were calculated using a vertical dimer from the mouse CaMKII-α hub, using an elastic network model (*Suhre and Sanejouand, 2004*). In the diagram on the left, displacement vectors corresponding to the lowest-frequency internal normal mode are shown by arrows. The structures shown in magenta and blue correspond to excursions of the mouse hub along this mode. In the middle, the structures of the mouse hub and the *S. rosetta* hub are compared, and the arrows show the displacement vectors between the two structures. The diagram on the right shows the structures of the mouse hub and the *S. rosetta* hub, and excursions of the mouse hub along the lowest-frequency internal normal mode (green). Note that the normal mode, calculated without reference to the *S. rosetta* structure, captures the essential features of the structural difference between the mouse and *S. rosetta* hub. (**B**) Comparison of the *S. rosetta* hub structure to instantaneous structures from a molecular dynamics trajectory for a decameric human hub assembly (the open decameric assembly is generated by removing a vertical dimer from the dodecameric hub ring; PDB code: 2UX0). Helices A to D in the *S. rosetta* hub are aligned to the corresponding helices in the instantaneous simulated structures. Three instantaneous structures from the trajectory, at 1, 7 and 41 ns, are shown. Note that the strands of the β-sheet in structures at 7 and 41 ns are closely aligned with the *S. rosetta* structure.

implemented in *elNémo* (*Bahar and Rader, 2005*; *Suhre and Sanejouand, 2004*) (see Materials and methods). The lowest-frequency internal normal mode corresponds to a change in the curvature of the β-sheet, such that one end of the β-sheet alters its distance from the helical layer, which stays relatively rigid. If the structure of the mouse hub domain is displaced along this normal mode, then the changes in structure recapitulate the essential differences between the structures of the subunits in different hub assemblies, including the *S. rosetta* spiral, to a remarkable extent (*Figure 11A*). Low-frequency normal modes are indicative of low-energy deformations of the protein

fold (*Bahar and Rader, 2005*), and so this suggests that the mammalian hub domains are intrinsically capable of the structural deformation that is required for the formation of an open spiral.

We had previously described molecular dynamics simulations in which the constraint of ring closure had been released by removing one dimer from a human dodecameric hub (*Stratton et al., 2014*). We had then carried out two simulations of this system, extending for 100 ns and 50 ns, respectively. Inspection of instantaneous structures sampled from these trajectories shows that the structures of individual hub subunits undergo transient excursions in which the curvature of the β sheet is close to that seen in the *S. rosetta* structure (*Figure 11B*). The hub does not go out of plane during these relatively short simulations. Presumably, adoption of the spiral form would require all subunits to change the curvature of the β-sheet in a correlated manner, which would require much longer simulation times.

Finally, we generated a model for a human hub in the spiral configuration by taking the *S. rosetta* spiral structure and replacing the sequence of each domain by the human one. We then initiated molecular dynamics trajectories from these modeled spiral structures (two replicates; 150 ns and 100 ns, respectively). The spiral geometry is stable in these trajectories, and the packing interactions at the hub interfaces remain intact (data not shown). This indicates that the human sequence is consistent with the spiral architecture seen in the *S. rosetta* structure. The *S. rosetta* spiral may represent a particularly extreme distortion of the hub assembly – a smaller spiral distortion in human CaMKII than seen in *S. rosetta* may suffice to allow the release and capture of dimer units without the steric hindrance that is a feature of the nearly-closed *N. vectensis* spiral.

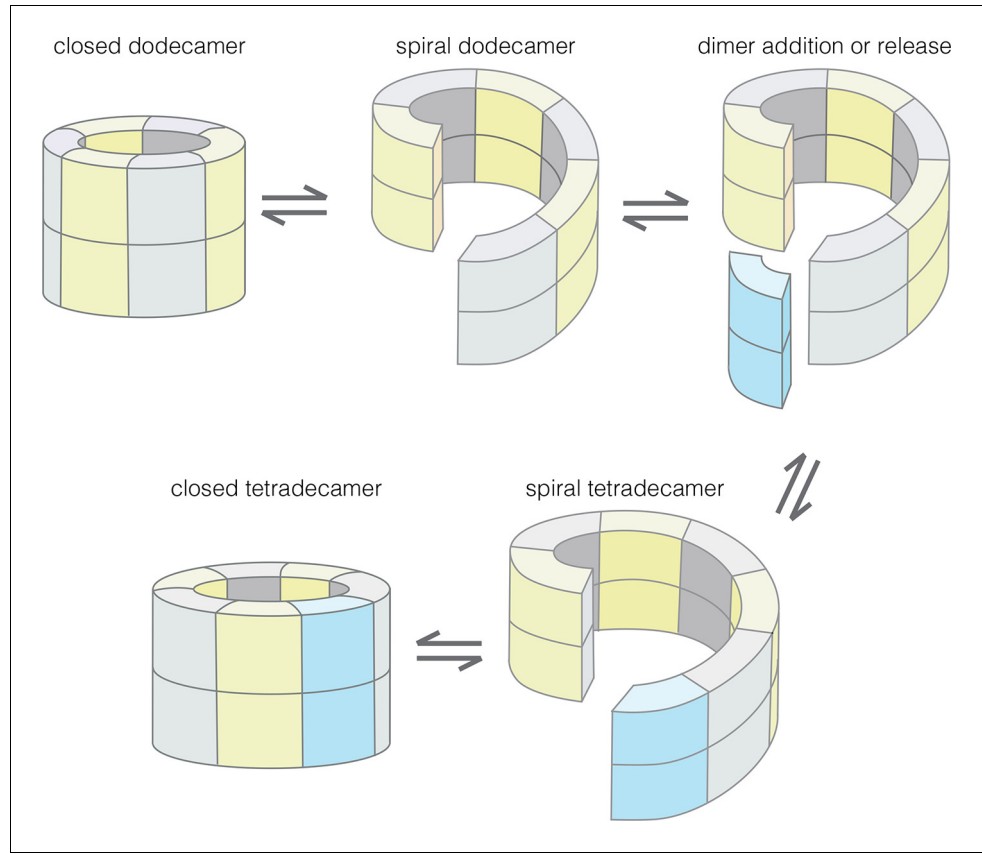

**Figure 12.** Model for the interconversion of dodecameric and tetradecameric assemblies by the CaMKII hub domain. This could potentially support an efficient subunit exchange mechanism that does not require complete disassembly and reassembly.

## Concluding remarks

Our results provide an understanding of how the structural integrity of CaMKII holoenzymes is altered by activation, thereby triggering subunit exchange. Subunit exchange in CaMKII requires a remarkable degree of architectural flexibility in the hub domain, which has long been an under-appreciated component of CaMKII. Although it gives the holoenzyme its distinctive shape and symmetry, the hub domain has neither the catalytic activity nor the sites of regulatory post-translational modification that has focused attention on the kinase domain and the autoinhibitory segment. We now show that the hub domain enables the CaMKII holoenzyme to switch between closed-ring dodecameric and tetradecameric states, and to potentially adopt ring-opened spiral configurations that can reversibly add and release dimer units to enable subunit exchange and interconversion between these states (*Figure 12*).

The plasticity in the hub domain presumably emerged as it diverged in evolution from a class of enzymes typified by the keto-steroid isomerases (*Hoelz et al., 2003*; *Wu et al., 1997*). The fold of these enzymes, which is also shared by binding proteins such as nuclear transport factor 2 (*Stewart et al., 1998*), creates an active site between the helical layer and the β sheet. As noted earlier, this cavity is deeper in CaMKII than in the enzymes to which it is related (*Hoelz et al., 2003*). The cavity in the CaMKII hub contains several highly conserved charged residues, including three arginines. The positioning of these charged residues in an invagination between the two layers of the domain is what makes the hub domain so flexible – the cavity between the layers is solvated, and therefore able to readily accommodate changes in shape.

The regulation of CaMKII activity by $Ca^{2+}$/CaM occurs through a mechanism that is common to other $Ca^{2+}$-regulated kinases – the autoinhibitory segment serves as a pseudosubstrate that occupies the substrate-binding channel, and it is released from the kinase domain by the binding of $Ca^{2+}$/CaM or a related protein (*Goldberg et al., 1996*; *Hu et al., 1994*; *Pearson et al., 1988*). Our data suggest that the CaM-binding element has an additional function, which is to dock at the interfaces between hub dimers, and thereby distort the hub assembly. Our data are consistent with the idea that the CaM-binding element induces the hub to transition from a closed-ring form to a ring-opened lock-washer configuration, which can release or capture dimer units. One attractive feature of this model, which is not directly tested here, is that subunit exchange need not involve complete disassembly and reassembly of the hub, which is likely to be a much less efficient process.

The structures of CaMKII homologs from two distantly diverged organisms, the sea anemone *N. vectensis* and the choanoflagellate *S. rosetta*, provided crucial insight into the likely mechanism of subunit exchange. CaMKII, like many signaling proteins, is so highly conserved in higher metazoans that alteration in structure due to sequence variation is rather limited. Recent advances in the sequencing of genomes of early-branching metazoans and their closest relatives gives us an unprecedented opportunity to understand the core mechanisms that underlie the functions of critical signaling proteins, such as CaMKII, that are common to all metazoans (*Richter and King, 2013*).

Activation-triggered subunit exchange in macromolecular assemblies has been reported only very rarely. One system in which such a process is thought to occur is the molecular clock formed by the KaiABC complex, which maintains diurnal rhythm in cyanobacteria (*Kageyama et al., 2006*; *Markson and O'Shea, 2009*). The exchange of histone subunits between existing nucleosomes and newly synthesized ones may be part of the mechanism by which epigenetic modifications are maintained during replication and transcription (*Campos et al., 2014*; *Xu et al., 2010*). A form of activation-dependent subunit exchange underlies the function of the Mad2 protein in controlling chromosome separation (*Musacchio, 2015*). Certain oligomeric enzymes undergo alterations in the stoichiometry and architecture of their quaternary structures in response to the binding of allosteric effectors, a process that involves disassembly and reassembly of the subunits (*Selwood and Jaffe, 2012*). Activation-triggered subunit exchange provides a general mechanism to synchronize the functional states of a population of molecules, so that changes initiated by an input signal can spread from a small number of activated assemblies to many. CaMKII functions at synapses and in neuromuscular junctions, and our findings explain how its activation triggers the release of dimers that can spread and maintain the effects of calcium spikes. We anticipate that the structural mechanism we have delineated will enable future experiments in which mutations designed to alter subunit exchange in CaMKII are correlated with changes in its many crucial signaling functions.

## Materials and methods

### Molecular biology

Full length CaMKII from *S. rosetta* and the CaMKII hub domains from *N. vectensis* were cloned from their respective cDNA libraries (*Fairclough et al., 2013*; *Putnam et al., 2007*) (*S. rosetta* cDNA provided by Nicole King, UC Berkeley and *N. vectensis* cDNA provided by Nathaniel Clarke & Christopher Lowe, Stanford University) using standard PCR techniques. These genes, as well as DNA corresponding to the human hub, were cloned into a pET-28 vector (Novagen), modified to contain a PreScission Protease (Pharmacia) site between the N-terminal 6-histidine tag and the coding sequence. Human full-length CaMKII-α and its variants as well as CaMKII-β were cloned into a pSMT-3 vector containing an N-terminal SUMO expression tag (LifeSensors, Malvern, PA). Mutants were generated using Quikchange protocol (Agilent Technologies, Santa Clara, CA) and the constructs with domain truncations were made using standard PCR techniques.

### Protein expression and purification

All human CaMKII constructs and their variants were expressed in *E. coli* and purified as described (*Chao et al., 2011*). Briefly, for the full length CaMKII variants, protein expression was done in Tuner (DE-3) pLysS cells that contained an additional plasmid for λ phosphatase production (*Chao et al., 2010*). For constructs that do not have a kinase domain, as well as for calmodulin, protein expression was carried out in BL21 cells. CaMKII-β was co-expressed in Rosetta2 (DE3) pLysS cells with λ phosphatase (expressed on a separate plasmid). Cells were induced by the addition of 1 mM isopropyl β-D-1-thiogalactopyranoside and grown overnight at 18°C. Cell pellets were resuspended in Buffer A (25 mM Tris, pH 8.5, 150 mM potassium chloride, 1 mM DTT, 50 mM imidazole, and 10% glycerol for human CaMKII variants and all hub domains including those from *S. rosetta* and *N. vectensis*; 25 mM Tris, pH 8.5, 600 mM sodium chloride, 1 mM DTT, 50 mM imidazole, and 10% glycerol for the *S. rosetta* CaMKII holoenzyme) and lysed using a cell disrupter. Filtered lysate was loaded on 5 mL Ni-NTA column, eluted with 0.5 M imidazole (0.76 M imidazole for human CaMKII-α hub), desalted using a HiPrep 26/10 desalting column into Buffer A with 10 mM imidazole, and cleaved with Ulp1 or PreScission protease (overnight at 4°C). The cleaved samples were loaded onto the Ni-NTA column and the flow through was loaded onto a Q-FF 5 ml column, and then eluted with a KCl (or NaCl in case of *S. rosetta* holoenzyme) gradient. Eluted proteins were purified further using a Superose 6 gel filtration column equilibrated in gel filtration buffer (25 mM Tris, pH 8.0, 150 mM KCl, 1.0 mM tris(2-carboxyethyl)phosphine [TCEP] and 10% glycerol for the human CaMKII variants and all hub domains including those from *S. rosetta* and *N. vectensis*; 50 mM BisTrisPropane, pH 7.2, 200 mM NaCl, 1.0 mM TCEP and 10% glycerol for the *S. rosetta* CaMKII holoenzyme; 2 mM DTT was added and no glycerol was used in the gel filtration buffer for the human CaMKII-α hub). One additional step was added for the human CaMKII-α hub crystallization construct only, where after the gel-filtration column, the protein was diluted into 20 mL of a solubilization buffer (4 M urea, 25 mM Tris, pH 8.0, 150 mM KCl, 1 mM TCEP, 5% glycerol). The protein was then dialyzed into a final buffer of 25 mM Tris, pH 8.0, 150 mM KCl, 1.7 mM urea, 1 mM DTT, 0.5 mM TCEP, and 5% glycerol.

Fractions with pure protein were pooled, concentrated and frozen at −80°C. All purification steps were carried out at 4°C and all columns were purchased from GE Healthcare (Piscataway, NJ).

### Native electrospray ionization mass spectrometry (ESI-MS)

Mass spectra were acquired on a quadrupole time-of-flight mass spectrometer (Q-TOF Premier, Waters, Milford, MA, USA) with the backing pressure increased to ∼6 mbar to promote ion desolvation of large protein complexes. Ions were formed by nanoelectrospray ionization from borosilicate capillaries (1.0 mm o.d./0.78 mm i.d, Sutter Instruments, Novato, CA, USA) that were pulled to a tip i.d. of ∼1 μm with a Flaming/Brown micropipette puller (Model P-87, Sutter Instruments, Novato, CA, USA). The tip of the capillary was held ∼8–10 mm from the mass spectrometer inlet, and nanoelectrospray was initiated by applying ∼1.2–1.5 kV relative to instrument ground to a platinum wire (0.127 mm diameter, Sigma, St. Louis, MO, USA) that is in contact with the sample solution. The instrument was calibrated with cesium iodide (CsI) clusters formed from solutions consisting of 20 mg/mL CsI in 70:30 Milli-Q water:2-propanol. Protein samples were buffer exchanged into 1 M

ammonium acetate, pH 7.2, via ion exchange spin columns (Bio-Spin 6, Bio-Rad Laboratories, Inc., Hercules, CA, USA) and then diluted to the desired concentration with this same buffer. Collision induced dissociation (CID) of protein complexes was performed by applying an accelerating voltage to ions in a collision cell containing argon gas at a pressure of 8 mbar. For non-dissociative conditions, this voltage was held at 5 V, where no complex fragmentation was observed, and for CID conditions, this voltage was increased until dissociation of the complexes was observed. Raw data were smoothed three times using the Waters MassLynx software Savitsky-Golay smoothing algorithm with a smoothing window of 50–100 $m/z$ (mass-to-charge ratio).

## Negative-stain electron microscopy

For negative-stain electron microscopy, a 5 µL sample of CaMKII-α protein (15 µg/mL) in 20 mM Tris pH 8.0, 150 mM KCl and 5% glycerol was placed on the continuous carbon side of a glow-discharged copper grid (Ted Pella, Redding, CA, USA), and the excess sample was removed by wicking with filter paper after 1 min incubation. The bound particles were stained by floating the grids on four consecutive 30 µL drops of 2% uranyl acetate solution and incubating each drop for 10 s. The excess stain was removed by blotting with filter paper and grids were air-dried.

Images of stained full-length CaMKII-α were recorded on a 4049x4096 pixel CMOS camera (TVIPS TemCam-F416) using the automated Leginon data collection software (*Suloway et al., 2005*). Samples were imaged using Tecnai 12 transmission electron microscope (FEI, Hillsboro, OR, USA) at 120 keV at a nominal magnification of 49,000 (2.18 Å calibrated pixel size at the specimen level) using a defocus range of -0.8 to 1.5 µm. All data were acquired under low dose conditions, allowing a dose at around 35e$^-$ / A$^2$.

The initial image processing and classification steps were performed using the Appion image-processing environment (*Lander et al., 2009*). Particles were first selected from micrographs using DoG Picker (*Voss et al., 2009*). The contrast transfer functions (CTFs) of the micrographs were estimated using the CTFFIND (*Mindell and Grigorieff, 2003*). CTF correction of the micrographs was performed by Wiener filter using ACE2 (*Mallick et al., 2005*). A total of 25,555 particles were extracted using a 176x176 pixel box size and binned by a factor of 2. Each particle was normalized to remove pixels whose values were above or below 4.5 σ of the mean pixel value using the XMIPP normalization program (*Scheres et al., 2008*). In order to remove incorrectly selected protein aggregates or other artifacts, particles whose mean intensity deviated too much from the mean value of the data set were removed. The remaining 25,485 particles were subjected to 2D iterative reference-free alignment and classification using a topology-representing network classification and IMAGIC multi-reference alignment (MRA) (*van Heel et al., 1996*; *Ogura et al., 2003*). The initial 200 2D class averages were manually inspected to check for classes that appeared to correspond to protein aggregates and contaminants. The 25,485 particles were subjected to another two rounds of MRA and classification, to produce the final 50 2D class averages. No symmetry operator was applied at any point in this analysis.

## Peptide binding and competition assays

For fluorescence polarization experiments, a peptide spanning the CaM-binding element of human CaMKII-α (Peptide A: [296]RRKLKGAILTTMLATR[311]C) was synthesized by David King at the HHMI mass spectrometry facility, UC Berkeley. Phosphorylated peptides, corresponding to the sequence [290]LKKFNARRKLKGAILTTMLA[309]C (phosphorylated at either Thr 305 or Thr 306; Peptides C and D) were obtained from Elim Biopharm (Hayward, CA). Other peptides corresponding to parts of the autoinhibitory segment of human CaMKII-α and variants thereof (Peptide B: [290]LKKFNARRKLKGAI LTTMLA[309], Peptide E: [290]LKKFNAERKLKGAILTTMLA[309], Peptide F: [290]LKKFNARRKLEGAI LTTMLA[309], Peptide G: [290]LKEFNAERKLEGAILTTMLA[309],) were a gift of Leta Nutt, St. Jude Children's Research Hospital. All peptides were purified by HPLC and the purity was assessed by HPLC and/or mass spectrometry.

Peptides A, C and D were labeled with BODIPY FL-maleimide (ThermoFisher Scientific, Waltham, MA). BODIPY FL-maleimide was dissolved in DMSO. This was added in 1.5 fold molar excess to 500 µM peptide solution (final concentration) in Tris buffer at pH 7.4 in the presence of 1 mM TCEP. The reaction mixture was incubated for ∼10 min at room temperature. The reaction time was optimized using analytical HPLC profiles using ∼15–20 µg of peptide samples from the labeling reaction

mixture. The BODIPY labeled peptides were purified using HPLC and characterized using mass spectrometry (data not shown).

The fluorescence polarization binding experiments were initiated by adding 15 µL of 2 nM BODIPY labeled peptides (buffer: 25 mM Tris at pH 8.0) to 15 µL of different concentrations of the human CaMKII-α hub domain (ranging from 0–1000 µM in 25 mM Tris at pH 8.0, 150 mM KCl, 10% glycerol and 0.02% Tween). For the competition assays, 10 µL of BODIPY labeled peptides at 2 nM were added to 10 µL of hub at 30 µM. This was followed by the addition of 10 µL of the unlabeled peptides in excess (concentration ranging between 0–500 µM in 25 mM Tris at pH 8.0) to these reaction mixtures. The fluorescence polarization was read from 20 µL of these reaction mixtures in opaque black 96-well plates using a Synergy H4 hybrid microplate reader with a 485/20 nm excitation filter, and a 528/20 nm emission filter. Error bars are calculated from the standard error of mean between replicates of experiments.

The competition assay requires us to add both the hub and the unlabeled peptide at concentrations that are close to the value of $K_D$. Under these conditions, the determination of the value of $K_D$ requires the use of an equation that is cubic in concentration (*Wang, 1995*). We made the simplifying assumption that the dissociation constants of the labeled and unlabeled peptides are the same, and took advantage of the fact that the concentration of the labeled peptide is very small compared to the hub concentration. Given these assumptions, the following equation (that is second-order in concentration) can be derived:

$$A = A_f + (A_0 - A_f) \times \frac{K_D + [P]_{tot}}{[P]_{tot}} \times \frac{([P]_{tot} + [L]_{tot} + K_D) - \sqrt{([P]_{tot} + [L]_{tot} + K_D)^2 - 4[P]_{tot}[L]_{tot}}}{2[L]_{tot}}$$

where, A is the observed fluorescence polarization (FP) value, $A_f$ is the FP value for free labeled ligand, $A_0$ is the FP value in the absence of unlabeled ligand, $K_D$ is the dissociation constant, $[P]_{tot}$ is the total protein concentration, and $[L]_{tot}$ is the total concentration of the unlabeled ligand. We determined the $K_D$ values by non-linear fitting using this equation.

## FRET experiments

To optimize labeling and the FRET signal, two surface-exposed cysteine residues in CaMKII-α (Cys 280 and Cys 289) were mutated to serine, and Asp 335 was mutated to cysteine (the numbering is according to PDB code: 3SOA), as described (*Stratton et al., 2014*). Similarly, in *S. rosetta* CaMKII, Cys 230 was mutated to a serine and Asp 367 was mutated to cysteine. For CaMKII-β, no mutations were made and the wild-type protein was labeled. Labeling with Alexa-488 and Alexa-594 was achieved as described (*Stratton et al., 2014*). Briefly, purified CaMKII variant was mixed with three to five-fold molar excess of Alexa Fluor $C_5$-maleimide dyes (Alexa-488 and Alexa-594, Life Technologies) over CaMKII subunit concentration. This was incubated for 3–4 hr at 25°C. Excess dye at the end of the labeling reactions was removed using PD-25 or PD-10 columns (GE healthcare) equilibrated with 25 mM Tris at pH 8.0, 150 mM KCl, 10% glycerol and 1 mM TCEP. Samples were concentrated using Amicon filters and dye incorporation was estimated using spectrophotometric analysis (Nanodrop, Thermo Scientific, DE), as described (*Stratton et al., 2014*). The percentage of labeled cysteine residues varied between different proteins (ranging from ∼30% to 85%).

For the FRET assays, labeled samples were mixed at a final concentration of ∼5 µM and incubated at 25°C or 37°C. At each time point, 15 or 25 µl from the mixed sample was removed and diluted to a final volume of 150 µl. An emission spectrum (500–700 nm) was acquired for each diluted sample excited at 490 nm using a Fluoromax-3 fluorometer (Horiba Scientific, Edison, NJ). The data were analyzed by calculating the FRET ratio (acceptor emission at 614 nm divided by donor emission at 510 nm).

## Analytical gel filtration chromatography

For analytical gel filtration chromatography, samples were loaded onto a Superose 6 10/300 column (10/300 GL; GE Healthcare) equilibrated with 25 mM Tris at pH 8.0, 150 mM KCl, 10% glycerol and 1 mM TCEP or 2 mM DTT, at a flow rate of 0.3–0.5 ml/min (Prominence UFLC, Shimadzu). Beta amylase (200 kDa), gamma globulin (158 kDa), alcohol dehydrogenase (150 kDa), bovine serum albumin (66 kDa), chicken ovalbumin (44 kDa), carbonic anhydrase (29 kDa), gamma phosphatase (25 kDa),

myoglobin (17 kDa), and cytochrome C (12.4 kDa) were used as molecular weight standards to calibrate the column (see *Figure 4—figure supplement 4* for the calibration curve). The standard proteins were obtained from Sigma Aldrich, Bio-Rad and New England Biolabs.

## Crystallization of CaMKII hub domains

The crystallization construct for the human CaMKII-α hub consisted of residues 345–475 (UNIPROT id: Q9UQM7). This sequence was preceded by an expression tag containing hexahistidine followed by a PreScission protease cleavage site (GSSHHHHHHSSGLEVLFQGPHM). This expression tag was left uncleaved for crystallization. The crystallization construct for the *N. vectensis* hub domains (CaMKII-B and CaMKII-A) included residues 335–476 (UNIPROT id: A7RF52) and residues 331–472 (UNIPROT id: A7T0H5) respectively. For *S. rosetta* CaMKII, the hub domain crystallization construct is comprised of residues 335–479 and the kinase domain construct ranged from residues 1–330 (UNIPROT id: F2UPG5). The kinase domain construct for *S. rosetta* had an additional 11 residues, containing a hexahistidine tag on the C-terminus (AAALEHHHHHH). The crystallization construct for the *N. vectensis* hub domains and that of *S. rosetta* were similar to that of the mouse CaMKII-α hub (PDB code: 1HKX), and included 12 C-terminal residues of the linker (9 C-terminal residues of the linker were included in the construct for 1HKX).

Crystals were grown at 22°C (except for the human CaMKII-α hub for which the trays were stored at 4°C for the first 24 hours before moving them to 22°C) using the sitting drop vapor diffusion technique, using the following compositions for the reservoir - human CaMKII-α hub: 35% (v/v) MPD, 0.1 M HEPES pH 7.3; *S. rosetta* hub: 0.2 M lithium sulphate, 0.1 M Tris pH 7.0, 2.0 M ammonium sulphate; *N. vectensis* CaMKII-B hub (pH 4.2): 2.5 M sodium chloride, 0.1 M acetate pH 4.2, and 0.25 M lithium sulphate; *N. vectensis* CaMKII-A hub (pH 7.0): 0.2 M potassium thiocyanate, 20% w/v PEG 3350; *S. rosetta* kinase: 1.8 M sodium phosphate monobasic monohydrate, potassium phosphate dibasic pH 6.9. The crystals were cryoprotected in 25–30% glycerol and X-ray diffraction data were collected at the Advanced Light Source using beamlines 8.2.1, 8.2.2 and 5.0.2 at 100 K.

## Structure determination and refinement

All CaMKII hub structures were solved by molecular replacement using Phaser (*McCoy et al., 2007*) and the structure of mouse CaMKII-α hub domain (PDB code: 1HKX) as the search model. The model used for the molecular replacement of the *S. rosetta* kinase domain was the *C. elegans* CaMKII-α kinase domain (PDB code: 2BDW). The model building was done using Coot (*Emsley et al., 2010*) and refinement was performed in Phenix (*Adams et al., 2010*). Model validation was performed using Molprobity (*Chen et al., 2010*). The details of the data collection and refinement statistics for each structure are given in *Supplementary file 1*.

For the *N. vectensis* pH 4.2 structure (CaMKII-B hub), there is a crystallographic axis of 2-fold symmetry that runs through the middle of the tetramer formed by the E-F and G-H dimers (see *Figure 6—figure supplement 1*). The left-handed lock-washer architecture allows 12 of the 14 subunits in the pH 4.2 structure to obey this axis of symmetry. The M-N dimer cannot obey the symmetry, because of the spiral geometry. The symmetry axis generates two copies of the M-N dimer. One copy is "joined" to the A-B dimer, and dislocated from the K-L dimer. The other copy is joined to the K-L dimer, but dislocated from the A-B dimer (the structural assemblies showing these two alternate conformations of the M-N dimer are available as *Supplementary file 2* and *3*). The presence of these two copies was apparent upon examination of difference electron density maps, calculated from the initial molecular replacement solution. The alternate conformations of the M-N dimer could be due to crystal twinning, or static disorder. A clear distinction cannot be made between the two at the resolution of the data. We treated this as static disorder, with 50% occupancy for each conformation of the dimeric unit. Alternating model refinement and rebuilding steps were performed using Phenix and Coot respectively. NCS restraints were imposed during refinement along with the use of the Translation-Libration-Screw-rotation model, as implemented in Phenix.

## Multi-angle light scattering (MALS)

For MALS studies on the *S. rosetta* hub, purified protein at ~3.2 mg/ml was injected into a Superdex 200 10/300 analytical SEC column equilibrated overnight in gel filtration buffer (25 mM Tris at pH

8.0, 150 mM KCl, 1 mm TCEP, and 10% glycerol). The chromatography system was coupled to an 18-angle light scattering detector (DAWN HELEOS-II) and a refractive index detector (Optilab T-rEX) (Wyatt Technology). Data were collected every second and the flow rate was set to 0.5 ml/min. Data analysis was carried out using the program ASTRA (Wyatt Technology). Monomeric bovine serum albumin (BSA; Sigma) was used for calibration of the light scattering detectors and data quality control. Measurement was carried out at 25°C.

## Molecular dynamics simulations

Molecular dynamics trajectories were generated using the Gromacs 5.1 package (*Berendsen et al., 1995*; *Páll et al., 2015*) and Amber14 (*Case et al., 2014*). The ff99SB-ILDN force field (*Lindorff-Larsen et al., 2010*) was used for all the calculations. All simulations were carried out in water using the TIP3P water model and appropriate counter ions ($Na^+$ and $Cl^-$) were added to neutralize the net charges. After initial energy minimization, the systems were subjected to 100–500 ps of constant number, volume and temperature (NVT) equilibration, during which the system was heated to 300K. This was followed by a short equilibration at constant number, pressure and temperature (NPT, 100–500 ps). The equilibration steps were performed with harmonic positional restraints on all protein or peptide atoms. For the simulation with the tetradecamer and dodecamer bound to the RRKLK motif-containing peptide, additional steps of NVT and NPT equilibrations were added with harmonic positional restraints on the protein but not on the peptide atoms. Finally, the production simulations were performed under NPT conditions, with the Berendsen and v-rescale thermostats in Amber14 and Gromacs 5.1.0 respectively, in the absence of positional restraints. Periodic boundary conditions were imposed, and particle-mesh Ewald summations were used for long-range electrostatics and the van der Waals cut-off was set at 1 nm. A time step of 2 fs was employed and the structures were stored every 2 ps.

## Normal mode analysis

The normal mode analysis was performed using the web based program *elNémo* and the structure of a dimer from the mouse CaMKII-α hub (PDB code: 1HKX) (*Suhre and Sanejouand, 2004*). Five low frequency modes were obtained using the default settings of the web server and visual inspection of the modes were carried out using Pymol.

## Acknowledgement

We thank members of the Kuriyan lab, especially Jeanine Amacher, Aaron Cantor and Qi Wang for helpful comments on the manuscript, Neel H Shah for helpful discussions, Gail Odom and Aaron Cantor for help with figure design, and Qi Wang for help with MALS experiments. We thank David King (HHMI, Mass Spectrometry, UC Berkeley) and Leta Nutt (St. Jude Children's Research Hospital) for generous assistance with synthesis of peptides. We thank Tony Iavarone (QB3) for mass spectrometry support. We thank Nicole King for insights into early metazoan evolution and for providing the cDNA libraries, and Yamuna Krishnan for critical comments on the manuscript.

## Additional information

### Competing interests

JK: Senior editor, *eLife.* The other authors declare that no competing interests exist.

### Funding

| Funder | Grant reference number | Author |
|---|---|---|
| Howard Hughes Medical Institute | | John Kuriyan |
| National Institutes of Health | R01GM101277 | Howard Schulman |
| Human Frontier Science Program | | Moitrayee Bhattacharyya |
| Jane Coffin Childs Memorial Fund for Medical Research | | Margaret M Stratton |

National Institutes of Health          R01GM097357          Evan R Williams

The funders had no role in study design, data collection and interpretation, or the decision to submit the work for publication.

## Author contributions

MB, MMS, Conception and design, Acquisition of data, Analysis and interpretation of data, Drafting or revising the article; CCG, EDM, YH, ACS, AE, YMC, NP, TB, Acquisition of data, Analysis and interpretation of data, Drafting or revising the article; PB, Drafting or revising the article, Contributed unpublished essential data or reagents; CLG, HS, ERW, Analysis and interpretation of data, Drafting or revising the article; JK, Conception and design, Analysis and interpretation of data, Drafting or revising the article

## Author ORCIDs

Christine L Gee, http://orcid.org/0000-0002-2632-6418
Tiago Barros, http://orcid.org/0000-0002-9807-7625
John Kuriyan, http://orcid.org/0000-0002-4414-5477

# Additional files

## Supplementary files

• Supplementary file 1. Supplementary Table 1 showing the details of the data collection and refinement statistics for each crystal structure.

• Supplementary file 2. PDB coordinates for the structural assembly of the *N. vectensis* CaMKII-B hub conformation, with the M-N dimer joined to the K-L dimer, but dislocated from the A-B dimer.

• Supplementary file 3. PDB coordinates for the structural assembly of the *N. vectensis* CaMKII-B hub conformation, with the M-N dimer joined to the A-B dimer, but dislocated from the K-L dimer.

## Major datasets

The following datasets were generated:

| Author(s) | Year | Dataset title | Dataset URL | Database, license, and accessibility information |
|---|---|---|---|---|
| Bhattacharyya M, Gee CL, Barros T, Kuriyan J | 2016 | Crystal structure of *S. rosetta* CaMKII hub | www.rcsb.org/pdb/explore/explore.do?structureId=5IG0 | Publicly available at the RCSB Protein Data Bank (accession no. 5IG0) |
| Bhattacharyya M, Gee CL, Barros T, Kuriyan J | 2016 | Crystal structure of *S. rosetta* CaMKII kinase domain | www.rcsb.org/pdb/explore/explore.do?structureId=5IG1 | Publicly available at the RCSB Protein Data Bank (accession no. 5IG1) |
| McSpadden E, Cao YM, Bhattacharyya M, Gee CL, Barros T, Kuriyan J | 2016 | Crystal structure of the hub domain of human CaMKII alpha | http://www.rcsb.org/pdb/explore/explore.do?structureId=5IG3 | Publicly available at the RCSB Protein Data Bank (accession no. 5IG3) |
| Bhattacharyya M, Pappireddi N, Gee CL, Barros T, Kuriyan J | 2016 | Crystal structure of *N. vectensis* CaMKII-A hub | http://www.rcsb.org/pdb/explore/explore.do?structureId=5IG4 | Publicly available at the RCSB Protein Data Bank (accession no. 5IG4) |
| Bhattacharyya M, Gee CL, Barros T, Kuriyan J | 2016 | Crystal structure of *N. vectensis* CaMKII-B hub at pH 4.2 | http://www.rcsb.org/pdb/explore/explore.do?structureId=5IG5 | Publicly available at the RCSB Protein Data Bank (accession no. 5IG5) |

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
