## [Decision Letter]

Thank you for submitting your work entitled "Molecular mechanism of activation-triggered subunit exchange in Ca^2+^/calmodulin-dependent protein kinase II" for consideration by *eLife*. Your article has been reviewed by three peer reviewers, one of whom is a member of our Board of Reviewing Editors. The evaluation has been overseen by Tony Hunter as the Senior Editor.

The reviewers have discussed the reviews with one another and the Reviewing Editor has drafted this decision to help you prepare a revised submission.

Summary:

The manuscript presents extensive structural, biochemical, and computational studies of CaMKII to support a particular model for subunit exchange, a critical aspect of CaMKII's biological function. In this model, the linker between the kinase domain and the hub domain, upon activation, is free to bind to the "horizontal" subunit interface in the hub. This changes the geometry in the interface subtly, but the change is sufficient to "crack open" the hub structure, allowing entry of a ("vertical") dimeric CaMKII unit. The model is nicely supported by a myriad of experiments including negative-stain electron microscopy, crystallography, FRET, normal-mode analysis, and molecular dynamics. Although other aspects of the CaMKII activation process, as they relate to the functional significance of subunit exchange, are not addressed here, this study represents a substantive advance in our understanding of CaMKII and merits publication in *eLife*.

Essential revisions:

The reviewers agree that the revisions may be limited to textual changes. Nonetheless, one reviewer (reviewer 1) also proposed a few experiments, which should be relatively speaking easily at-hand, which the authors may elect to carry out to deepen their understanding of the mechanism of subunit exchange, and in case they found the suggested experiments reasonable, informative, and technically feasible.

Specifically, reviewer 1 writes:

"The main tenet of the model proposed by the authors is that activation regulates subunit exchange and that this might be relevant for the spread of activation signals. Missing from the work is a demonstration that the mechanism is relevant for the function of CaMKII in a functional context. This concern may be mitigated if the authors opted to carry out a few additional experiments.

Specifically, The FRET assay developed by the authors appears to be ideally suited to ask the question whether subunits from activated oligomers can penetrate into inactive oligomers. For instance, can dimers from oligomers of the linker-hub construct penetrate oligomers of the hub-only constructs? The reason why this question is important is that if ring destabilization required to "accept" subunits from active oligomers also requires destabilization mediated by activation, the purpose of the exchange would remain unclear. The work presented here does not contribute to the clarification of this important point.

Also, is the process inhibited by the isolated kinase domain or by addition of CaM? And is it accelerated by phosphorylation with active kinase? Given that the emergence of the FRET signal can be followed as a function of time, can the authors clarify whether phosphorylation of the linker changes the kinetics of this process (even if it does not modify its thermodynamics)? These experiments ought to be relatively easy to perform and would significantly extend the characterization of the mechanism."

---

## [Author Response]

*Essential revisions:*

*The reviewers agree that the revisions may be limited to textual changes. Nonetheless, one reviewer (reviewer 1) also proposed a few experiments, which should be relatively speaking easily at-hand, which the authors may elect to carry out to deepen their understanding of the mechanism of subunit exchange, and in case they found the suggested experiments reasonable, informative, and technically feasible. Specifically, reviewer 1 writes: "The main tenet of the model proposed by the authors is that activation regulates subunit exchange and that this might be relevant for the spread of activation signals. Missing from the work is a demonstration that the mechanism is relevant for the function of CaMKII in a functional context. This concern may be mitigated if the authors opted to carry out a few additional experiments. Specifically, The FRET assay developed by the authors appears to be ideally suited to ask the question whether subunits from activated oligomers can penetrate into inactive oligomers. For instance, can dimers from oligomers of the linker-hub construct penetrate oligomers of the hub-only constructs? The reason why this question is important is that if ring destabilization required to "accept" subunits from active oligomers also requires destabilization mediated by activation, the purpose of the exchange would remain unclear. The work presented here does not contribute to the clarification of this important point.*

The reviewer’s query concerns the mechanism by which activated CaMKII can exchange subunits with unactivated CaMKII, a phenomenon that we had reported in our earlier paper (Stratton et al. 2014). We agree that this is an issue that is not addressed in the present manuscript. Specifically, the reviewer asks whether a construct containing the hub and the CaM-binding element (a portion of the autoinhibitory segment) can “break” hub assemblies that do not contain the CaM-binding element. We have done new experiments to address this issue, and find that a hub assembly bearing the CaM-binding element, but no kinase domains, cannot exchange with a “naked” hub construct.

We have now carried out the following subunit exchange experiments:

i) Mixing hub alone (i.e., no linker or CaM-binding element) with hub-△kinase;

ii) Hub alone with hub-linker (a construct in which the kinase domain and the autoinhibitory segments have been deleted);

iii) Hub-linker with hub-linker.

We observe that none of the combinations exhibit significant subunit exchange and we conclude that it is necessary for the CaM-binding element to be present for subunit exchange to occur. That is, a hub bearing the autoinhibitory segment cannot “break” a hub lacking the autoinhibitory segment. These results have been summarized in Figure 4—figure supplement 1 (figure is now cited in paragraph three, subheading “Deleting the kinase domain of CaMKII results in spontaneous subunit exchange”) in the revised manuscript. As reported in our earlier *eLife* paper (2014), activated CaMKII holoenzyme can exchange with unactivated CaMKII. In the present manuscript, we show that hub-△kinase exchanges with hub-△kinase. Thus, the presence of both the kinase domain and the autoinhibitory segment is necessary for activated species to exchange with unactivated species.

We don’t fully understand whether this effect is due to kinase activity or whether the kinase can release regulatory segments from unactivated holoenzymes by a displacement mechanism that we have demonstrated previously (Chao et al. 2010). This requires further study, and although we agree with the reviewer that it is conceptually important, we have concluded that the level of additional experimentation required is beyond the scope of our current work.

*Also, is the process inhibited by the isolated kinase domain or by addition of CaM?*

We presume that the reviewer is referring to subunit exchange in hub-△kinase, where neither CaM nor the kinase domain is present. We anticipate that the process should be inhibited by addition of the kinase domain or CaM. We report below that the autoinhibitory segment is a strong inhibitor of the kinase domain in trans, which implies that free kinase domain should inhibit the process. Our new experiments (see above) show that a construct containing the hub and the linker, without the CaM-binding element, does not exchange subunits, and so CaM should inhibit the process in the absence of phosphorylation. We agree that it would be interesting to study the effect of CaM on the exchange process, but do not feel that such experiments would provide a critical conceptual advance.

*And is it accelerated by phosphorylation with active kinase? Given that the emergence of the FRET signal can be followed as a function of time, can the authors clarify whether phosphorylation of the linker changes the kinetics of this process (even if it does not modify its thermodynamics)? These experiments ought to be relatively easy to perform and would significantly extend the characterization of the mechanism."*

We appreciate what the reviewer is getting at: does phosphorylation potentiate the ability of the autoinhibitory segment (including the CaM-binding element) to mediate exchange? We had, indeed, thought that this would be an important experiment to do, but quickly discovered a problem intrinsic to the experiment. The free autoinhibitory segment is a potent inhibitor of the kinase domain. We have now mentioned this point in paragraph two, subheading “The CaM-binding element of CaMKII binds to the hub with micromolar affinity” of the revised manuscript. Shown below are kinetic traces from a coupled-kinase assay, in which the kinase domain (without the autoinhibitory segment) is used as an enzyme to phosphorylate either a synthetic substrate (syntide) or peptide derived from the CaM-binding element. The assay monitors the depletion of ATP via changes in the concentration of NADH (Barker et al. 1995; Chao et al. 2010). Note that with syntide as a substrate, there is rapid diminution of signal until ATP is exhausted. Addition of CaM-binding element peptide as a substrate leads to no activity.

This result, which is consistent with the published literature on CaMKII activity, shows that phosphorylation in the holoenzyme is presumably mediated by transient dynamics of the autoinhibitory segment when it is presented to an enzyme at the high local concentrations of the subunits in the holoenzyme. When the autoinhibitory segment is presented in trans the inhibitory binding mode dominates, and we presume that phosphorylation occurs too rarely to be detected. This is an unfortunate result in terms of doing the experiment that the reviewer has suggested. We have not thought it worthwhile, at this point, to explore the use of unrelated kinases to carry out the phosphorylation, since we do not know what such a kinase might be. Note that our previous work (Stratton et al. 2014) indicated that phosphorylation is important, because mutation of either Thr 305 or Thr 306 reduced subunit exchange. We regret that the parameters of the system do not allow us to do the proposed experiment.